# Better Supervisory Signals
# by Observing Learning Paths

**Yi Ren**
UBC
renyi.joshua@gmail.com

**Shangmin Guo**
University of Edinburgh
s.guo@ed.ac.uk

**Danica J. Sutherland**
UBC and Amii
dsuth@cs.ubc.ca

## Abstract

Better-supervised models might have better performance. In this paper, we first clarify what makes for good supervision for a classification problem, and then explain two existing label refining methods, label smoothing and knowledge distillation, in terms of our proposed criterion. To further answer why and how better supervision emerges, we observe the learning path, i.e., the trajectory of the model's predictions during training, for each training sample. We find that the model can spontaneously refine "bad" labels through a "zig-zag" learning path, which occurs on both toy and real datasets. Observing the learning path not only provides a new perspective for understanding knowledge distillation, overfitting, and learning dynamics, but also reveals that the supervisory signal of a teacher network can be very unstable near the best points in training on real tasks. Inspired by this, we propose a new knowledge distillation scheme, Filter-KD, which improves downstream classification performance in various settings.

## 1 Introduction

In multi-class classification problems, we usually supervise our model with "one-hot" labels: label vectors $y$ which have $y_i = 1$ for one $i$, and $0$ for all other dimensions. Over time, however, it has gradually become clear that this "default" setup is not always the best choice in practice, in that other schemes can yield better performance on held-out test sets. One such alternative is to summarize a distribution of human annotations, as Peterson et al. (2019) did for CIFAR10. An alternative approach is label smoothing (e.g. Szegedy et al., 2016), mixing between a one-hot label and the uniform distribution. Knowledge distillation (KD), first training a teacher network on the training set and then a student network on the teacher's output probabilities, was originally proposed for model compression (Hinton et al., 2015) but can also be thought of as refining the supervision signal: it provides "soft" teacher outputs rather than hard labels to the student.

Knowledge distillation is promising because it requires no additional annotation effort, but – unlike label smoothing – can still provide sample-specific refinement. Perhaps surprisingly, knowledge distillation can improve student performance even when the teacher is of *exactly the same form as the student* and trained on the same data; this is known as self-distillation (Furlanello et al., 2018; Zhang et al., 2019). There have been many recent attempts to explain knowledge distillation and specifically self-distillation (e.g. Menon et al., 2021; Allen-Zhu & Li, 2020; Tang et al., 2020), from both optimization and supervision perspectives. We focus on the latter area, where it is usually claimed that the teacher provides useful "dark knowledge" to the student through its labels.

Inspired by this line of work, we further explore why and how this improved supervisory signal emerges during the teacher's one-hot training. Specifically, we first clarify that given any input sample $\mathbf{x}$, good supervision signals should be close (in $L_2$ distance) to the ground truth categorical distribution, i.e., $p^*(y \mid \mathbf{x})$. We then show that a neural network (NN) can automatically refine "bad labels", those where $p^*(y \mid \mathbf{x})$ is far from the training set's one-hot vector.[1] During one-hot training, the model prediction on such a sample first moves towards $p^*(y \mid \mathbf{x})$, and then slowly converges to its supervisory label, following a "zig-zag" pattern. A well-trained teacher, one that does not overfit

---

[1] This might be because $\mathbf{x}$ is ambiguous (perhaps $p^*(y \mid \mathbf{x})$ is flat, or we simply got a sample from a less-likely class), or because the one-hot label has been corrupted through label noise or otherwise is "wrong."

to particular training labels, can thus provide supervisory signals closer to $p^*(y \mid \mathbf{x})$. We justify analytically (Section 3.3) that this pattern is common in gradient descent training. Our explanations cause us to recognize that this signal can be better identified by taking a moving average of the teacher's prediction, an algorithm we term Filter-KD. This approach yields better supervision and hence better downstream performance, especially when there are many bad labels.

*After completing this work, we became aware of an earlier paper (Liu et al., 2020) studying almost the same problem in a similar way. We discuss differences between the papers throughout.*

## 2 SUPERVISION INFLUENCES GENERALIZATION

We begin by clarifying how the choice of supervisory signal affects the learned model.

### 2.1 CHOICES OF SUPERVISION SIGNAL

In $K$-way classification, our goal is to learn a mapping $f : \mathcal{X} \to \Delta^K$ that can minimize the risk

$$R(f) \triangleq \mathop{\mathbb{E}}_{(\mathbf{x},y)\sim\mathbb{P}}[L(y, f(\mathbf{x}))] = \int p(\mathbf{x})\, p(y|\mathbf{x})\, L(y, f(\mathbf{x}))\, \mathrm{d}\mathbf{x}\, \mathrm{d}y, \tag{1}$$

Here the label $y$ is an integer ranging from 1 to $K$, and $f$ gives predictions in the probability simplex $\Delta^K$ (a nonnegative vector of length $K$ summing to one); $L(y, f(\mathbf{x}))$ is the loss function, e.g. cross-entropy or square loss. The input signal $\mathbf{x}$ is usually high dimensional, e.g., an image or a sequence of word embeddings. The joint distribution of $(\mathbf{x}, y)$ is $\mathbb{P}$, whose density[2] can be written as $p(\mathbf{x})p(y|\mathbf{x})$. In practice, as $\mathbb{P}$ is unknown, we instead (approximately) minimize the empirical risk

$$R_{\mathrm{emp}}(f, \mathcal{D}) \triangleq \sum_{n=1}^{N}\sum_{k=1}^{K}\frac{1}{N}\,\mathbb{1}(y_n = k)\, L(k, f(\mathbf{x}_n)) = \sum_{n=1}^{N}\frac{1}{N}\,\mathbf{e}_{y_n}^{\mathsf{T}}\mathbf{L}(f(\mathbf{x}_n)), \tag{2}$$

where $\mathbb{1}(y_n = k)$ is an indicator function which equals 1 if $y_n = k$ or 0 otherwise, and $\mathbf{e}_{y_n} \in \{0,1\}^K$ is its one-hot vector form. $\mathbf{L}(f(\mathbf{x}_n)) = (L(1, f(\mathbf{x}_n)), \dots, L(K, f(\mathbf{x}_n))) \in \mathbb{R}^K$ is the loss for each possible label. In $R_{\mathrm{emp}}$, the $N$ training pairs $\mathcal{D} \triangleq \{(\mathbf{x}_n, y_n)\}_{n=1}^{N}$ are sampled i.i.d. from $\mathbb{P}$.

Comparing (2) to (1), we can see $p(\mathbf{x})$ is approximated by an uniform distribution over the samples, which is reasonable. However, using an indicator function (i.e., one-hot distribution) to approximate $p(y \mid \mathbf{x})$ bears more consideration. For example, if a data point $\mathbf{x}$ is quite vague and its true $p(y|\mathbf{x})$ is flat or multimodal, we might hope to see $\mathbf{x}$ multiple times with different label $y$ during training. But actually, most datasets have only one copy of each $\mathbf{x}$, so we only ever see one corresponding $\mathbf{e}_y$. Although $R_{\mathrm{emp}}$ is an unbiased estimator for $R$, if we used a better (e.g. lower-variance) estimate of $p(y \mid \mathbf{x})$, we could get a better estimate for $R$ and thus, hopefully, better generalization.

Specifically, suppose we were provided a "target" distribution $p_{\mathrm{tar}}(y \mid \mathbf{x})$ (written in vector form as $\mathbf{p}_{\mathrm{tar}}(\mathbf{x})$) for each training point $x$, as $\mathcal{D}' = \{(\mathbf{x}_n, \mathbf{p}_{\mathrm{tar}}(\mathbf{x}_n))\}_{n=1}^{N}$. Then we could use

$$R_{\mathrm{tar}}(f, \mathcal{D}') \triangleq \sum_{n=1}^{N}\sum_{k=1}^{K}\frac{1}{N}\, p_{\mathrm{tar}}(y_n = k \mid \mathbf{x}_n)\, L(k, f(x_n)) = \sum_{n=1}^{N}\frac{1}{N}\,\mathbf{p}_{\mathrm{tar}}(\mathbf{x}_n)^{\mathsf{T}}\mathbf{L}(f(\mathbf{x}_n)). \tag{3}$$

Standard training with $R_{\mathrm{emp}}$ is a special case of $R_{\mathrm{tar}}$, using $\mathbf{p}_{\mathrm{tar}}(\mathbf{x}_n) = \mathbf{e}_{y_n}$. The CIFAR10H dataset (Peterson et al., 2019) is one attempt at a different $\mathbf{p}_{\mathrm{tar}}$, using multiple human annotators to estimate $\mathbf{p}_{\mathrm{tar}}$. Label smoothing (e.g. Szegedy et al., 2016) sets $\mathbf{p}_{\mathrm{tar}}$ to a convex combination of $\mathbf{e}_y$ and the constant vector $\frac{1}{K}\mathbf{1}$. In knowledge distillation (KD; Hinton et al., 2015), a teacher is first trained on $\mathcal{D}$, then a student learns from $\mathcal{D}'$ with $\mathbf{p}_{\mathrm{tar}}$ based on the teacher's outputs. All three approaches yield improvements over standard training with $R_{\mathrm{emp}}$.

---

[2]Because we are working with classification problems, we use densities with respect to a product of some arbitrary measure on $\mathbf{x}$ (probably Lebesgue) with counting measure on $y$, and assume that these densities exist for notational convenience. None of our arguments will depend on the choice of base measure.

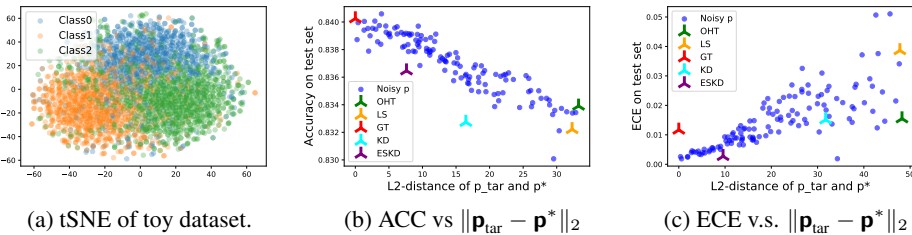

(a) tSNE of toy dataset.     (b) ACC vs $\|\mathbf{p}_{tar} - \mathbf{p}^*\|_2$     (c) ECE v.s. $\|\mathbf{p}_{tar} - \mathbf{p}^*\|_2$

Figure 1: Experiments on a toy dataset when learning from different $\mathbf{p}_{tar}$. In (b-c), the horizontal axis represents $\|\mathbf{p}_{tar} - \mathbf{p}^*\|_2$, and the vertical axis is the generalization performance. OHT means one-hot training (on $R_{emp}$), LS means label smoothing, GT means ground truth training with $\mathbf{p}^*$, KD is knowledge distillation, and ESKD is early-stopped KD. The Spearman correlation coefficient for results in (b) is -0.930 with p-value $1.9 \times 10^{-53}$; for (c) is 0.895 with p-value $2.7 \times 10^{-43}$.

## 2.2 Measuring the Quality of Supervision

Choosing a different $\mathbf{p}_{tar}$, then, can lead to a better final model. Can we characterize which $\mathbf{p}_{tar}$ will do well? We propose the following, as a general trend.

**Hypothesis 1.** *Suppose we train a model supervised by $\mathbf{p}_{tar}$, that is, we minimize $R_{tar}(f, \mathcal{D}')$. Then, smaller average $L_2$ distance between $\mathbf{p}_{tar}$ and the ground truth $\mathbf{p}^*$ on these samples, i.e. small $\mathbb{E}_{\mathbf{x}}[\|\mathbf{p}_{tar}(\mathbf{x}) - \mathbf{p}^*(\mathbf{x})\|_2]$, will in general lead to better generalization performance.*

This hypothesis is suggested by Proposition 3 of Menon et al. (2021), which shows (tracking constants omitted in their proof) that for any predictor $f$ and loss bounded as $L(y, \hat{y}) \leq \ell$,

$$\mathbb{E}_{\mathcal{D}'}\left[(R_{tar}(f, \mathcal{D}') - R(f))^2\right] \leq \frac{1}{N} \mathrm{Var}_{\mathbf{x}}\left[\mathbf{p}_{tar}^{\mathsf{T}} \mathbf{L}(f(\mathbf{x}))\right] + \ell^2 K \left(\mathbb{E}_{\mathbf{x}}\|\mathbf{p}_{tar}(\mathbf{x}) - \mathbf{p}^*(\mathbf{x})\|_2\right)^2. \quad (4)$$

When $N$ is large, the second term will dominate the right-hand side, implying smaller average $\|\mathbf{p}_{tar} - \mathbf{p}^*\|$ will lead to $R_{tar}$ being a better approximation of the true risk $R$; minimizing it should then lead to a better learned model. This suggests that the quality of the supervision signal can be roughly measured by its $L_2$ distance to the ground truth $\mathbf{p}^*$. Appendix B slightly generalizes the result of Menon et al. with bounds based on total variation ($L_1$) and KL divergences; we focus on the $L_2$ version here for simplicity.

To further support this hypothesis, we conduct experiments on a synthetic Gaussian problem (Figure 1 (a); details in Appendix C), where we can easily calculate $p^*(y \mid \mathbf{x})$ for each sample. We first generate several different $\mathbf{p}_{tar}$ by adding noise[3] to the ground truth $\mathbf{p}^*$, then train simple 3-layer NNs under that supervision. We also show five baselines: one-hot training (OHT), label smoothing (LS), KD, early-stopped KD (ESKD), and ground truth (GT) supervision (using $\mathbf{p}^*$). KD refers to a teacher trained to convergence, while ESKD uses a teacher stopped early based on validation accuracy. We early-stop the student's training in all settings. From Figure 1 (b-c), it is clear that smaller $\|\mathbf{p}_{tar} - \mathbf{p}^*\|_2$ leads to better generalization performance, as measured either by accuracy (ACC) or expected calibration error (ECE)[4] on a held-out test set. Appendix C has more detailed results.

## 3 Insights from the Learning Path

In the toy example of Section 2, we see that ESKD outperforms other baselines in accuracy by a substantial margin (and all baselines are roughly tied in ECE). We expect that supervision with smaller $\|\mathbf{p}_{tar} - \mathbf{p}^*\|_2$ leads to better generalization performance, but it is not clear how better $\mathbf{p}_{tar}$ emerges from when the teacher in ESKD is trained using one-hot labels. This section will answer this, by observing the learning paths of training samples.

---

[3]Whenever we mention adding noise to $\mathbf{p}^*$, we mean we add independent noise to each dimension, and then re-normalize it to be a distribution. Large noise can thus flip the "correct" label.

[4]ECE measures the calibration of a model (Guo et al., 2017). Briefly, lower ECE means the model's confidence in its predictions is more accurate. See Appendix A for details.

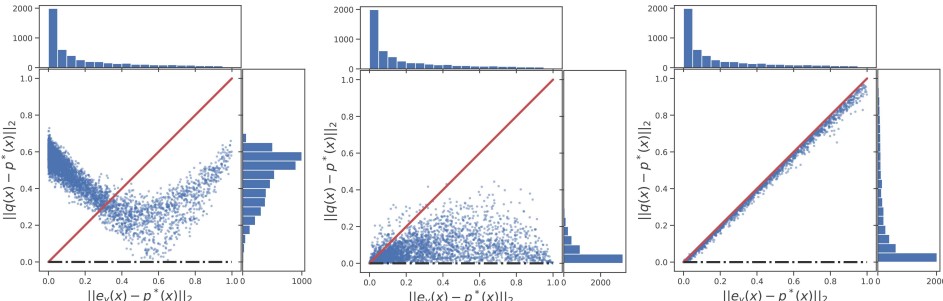

Figure 2: Normalized (divided by $\sqrt{2}$) distance between output distribution $\mathbf{q}$ and $\mathbf{p}^*$ during the one-hot training in different stages (left to right: initial, early stop, convergence). In these figures, $\mathbb{E}_\mathbf{x} \|\mathbf{q}(\mathbf{x}) - \mathbf{p}^*(\mathbf{x})\|_2$ of Hypothesis 1 is the mean height of all points in the figure. We provide more results about the NNs trained under different supervisions using this fashion in Appendix F.

### 3.1 PAY MORE ATTENTION TO HARDER SAMPLES

For a finer-grained understanding of early stopping the teacher, we would like to better understand how the teacher's predictions evolve in training. Assuming Hypothesis 1, the main factor is $\mathbb{E}_\mathbf{x}\|\mathbf{q}(\mathbf{x}) - \mathbf{p}^*(\mathbf{x})\|_2$, where $\mathbf{q}$ is the teacher's output probability distribution. We expect, though, that this term will vary for different $\mathbf{x}$, in part because some samples are simply more difficult to learn. As a proxy for this, we define **base difficulty** as $\|\mathbf{e}_y - \mathbf{p}^*(\mathbf{x})\|_2$, which is large if:

- $\mathbf{x}$ is ambiguous: $\mathbf{p}^*$ has several large components, so there is no one-hot label near $\mathbf{p}^*$.
- $\mathbf{x}$ is not very ambiguous (there is a one-hot label near $\mathbf{p}^*$), but the sample was "unlucky" and drew $y$ from a low-probability class.

Figure 2 shows these two quantities at three points in training: initialization, the point where ESKD's teacher stops, and convergence. At initialization, most points[5] have large $\|\mathbf{q}(\mathbf{x}) - \mathbf{p}^*(\mathbf{x})\|_2$. By the point of early stopping, most $\mathbf{q}(\mathbf{x})$ values are roughly near $\mathbf{p}^*$. At convergence, however, $\mathbf{q}(\mathbf{x}) \approx \mathbf{e}_y$, as the classifier has nearly memorized the training inputs, leading to a diagonal line in the plot.

It is clear the biggest contributors to $\mathbb{E}_\mathbf{x}\|\mathbf{q}(\mathbf{x}) - \mathbf{p}^*(\mathbf{x})\|_2$ when training a model to convergence are the points with high base difficulty. Per Hypothesis 1, these are the points we should most focus on.

### 3.2 LEARNING PATH OF DIFFERENT SAMPLES

To better understand how $\mathbf{q}$ changes for each sample, we track the model's outputs on all training samples at each training step. Figure 3 shows four samples with different base difficulty, with the vectors of three probabilities plotted as points on the simplex (details in Appendix A).

The two easy samples very quickly move to the correct location near $\mathbf{e}_y$ (as indicated by the light color until reaching the corner). The medium sample takes a less direct route, drifting off slightly towards $\mathbf{p}^*$, but still directly approaches $\mathbf{e}_y$. The hard sample, however, does something very different: it first approaches $\mathbf{p}^*$, but then veers off towards $\mathbf{e}_y$, giving a "zig-zag" path. In both the medium and hard cases, there seems to be some "unknown force" dragging the learning path towards $\mathbf{p}^*$; in both cases, the early stopping point is not quite perfect, but is noticeably closer to $\mathbf{p}^*$ than the final converged point near $\mathbf{e}_y$. In other words, during one-hot training, the NNs can *spontaneously refine the "bad labels."* Under Hypothesis 1, this partly explains the superior performance of ESKD to KD with a converged teacher.[6]

These four points are quite representative of all samples under different toy-dataset settings. In Appendix E, we also define a "zig-zagness score" to numerically summarize the learning path shape, and show it is closely correlated to base difficulty.

---

[5]The curve structure is expected: points with $\mathbf{p}^* \approx (\frac{1}{3}, \frac{1}{3}, \frac{1}{3})$ are near the middle of the base difficulty range, and all points are initialized with fairly ambiguous predictions $\mathbf{q}$.

[6]KD's practical success is also related to the temperature and optimization effects, among others; better supervision is not the whole story. We discuss the effect of various hyperparameters in Appendix D.

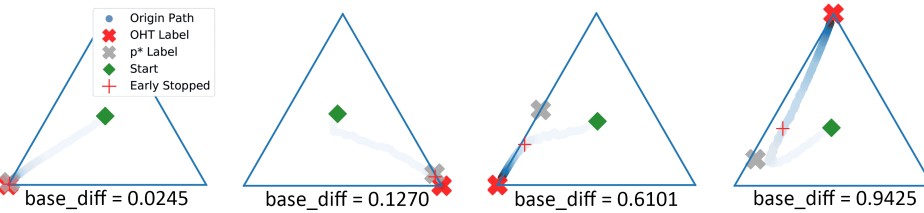

Figure 3: Learning path of samples with different base difficulty. Corners correspond to one-hot vectors. Colors represent training time: transparent at initialization, dark blue at the end of training.

### 3.3 EXPLANATION OF PATTERNS IN THE LEARNING PATH

The following decomposition will help explain the "unknown force" pushing $\mathbf{q}$ towards $\mathbf{p}^*$.

**Proposition 1.** *Let $\mathbf{z}^t(\mathbf{x}) \triangleq f(\mathbf{w}^t, \mathbf{x})$ denote the network output logits with parameters $\mathbf{w}^t$, and $\mathbf{q}^t(\mathbf{x}) = \mathrm{Softmax}(\mathbf{z}^t(\mathbf{x}))$ the probabilities. Let $\mathbf{w}^{t+1} \triangleq \mathbf{w}^t - \eta \nabla_{\mathbf{w}} \left(\mathbf{p}_{tar}(\mathbf{x}_u)^\mathsf{T} \mathbf{L}(\mathbf{q}^t(\mathbf{x}_u))\right)$ be the result of applying one step of SGD to $\mathbf{w}^t$ using the data point $(\mathbf{x}_u, \mathbf{p}_{tar}(\mathbf{x}_u))$ with learning rate $\eta$. Then the change in network predictions for a particular sample $\mathbf{x}_o$ is*

$$\mathbf{q}^{t+1}(\mathbf{x}_o) - \mathbf{q}^t(\mathbf{x}_o) = \eta \, \mathcal{A}^t(\mathbf{x}_o) \, \mathcal{K}^t(\mathbf{x}_o, \mathbf{x}_u) \, \left(\mathbf{p}_{tar}(\mathbf{x}_u) - \mathbf{q}^t(\mathbf{x}_u)\right) + \mathcal{O}(\eta^2 \|\nabla_{\mathbf{w}}\mathbf{z}(\mathbf{x}_u)\|_{\mathrm{op}}^2),$$

*where $\mathcal{A}^t(\mathbf{x}_o) = \nabla_{\mathbf{z}}\mathbf{q}^t(\mathbf{x}_o)$ and $\mathcal{K}^t(\mathbf{x}_o, \mathbf{x}_u) = (\nabla_{\mathbf{w}}\mathbf{z}(\mathbf{x}_o)|_{\mathbf{w}^t}) (\nabla_{\mathbf{w}}\mathbf{z}(\mathbf{x}_u)|_{\mathbf{w}^t})^\mathsf{T}$ are $K \times K$ matrices.*

The matrix $\mathcal{K}^t$ is the *empirical neural tangent kernel*, which we can think of roughly as a notion of "similarity" between $\mathbf{x}_o$ and $\mathbf{x}_u$ based on the network's representation at time $t$, which can change during training in potentially complex ways. In very wide networks, though, $\mathcal{K}^t$ is nearly invariant throughout training and nearly independent of the initialization (Jacot et al., 2018; Arora et al., 2019). The gradient norm can be controlled by gradient clipping, or bounded with standard SGD analyses when optimization doesn't diverge. Appendix G has more details and the proof.

Figure 4 shows the learning path of a hard sample during one-hot training, say $\mathbf{x}_o$, where the label $\mathbf{e}_{y_o}$ is far from $\mathbf{p}^*(\mathbf{x}_o)$. In each epoch, $\mathbf{w}$ will receive $N - 1$ updates based on the loss of $\mathbf{x}_u \neq \mathbf{x}_o$ (the small blue path), and one update based on $\mathbf{x}_o$ (the big red path). At any time $t$, "dissimilar" samples will have small $\mathcal{K}^t(\mathbf{x}_o, \mathbf{x}_u)$ (as measured, e.g., by its trace), and hence only slightly affect the predicted label for $\mathbf{x}_o$. Similar samples will have large $\mathcal{K}^t(\mathbf{x}_o, \mathbf{x}_u)$, and hence affect its updates much more; because $\mathbf{p}^*$ is hopefully similar for similar $\mathbf{x}$ values, it is reasonable to expect that the mean of $\mathbf{e}_y$ for data points with similar $\mathbf{x}$ will be close to $\mathbf{p}^*(\mathbf{x}_o)$. Thus the net effect of updates for $\mathbf{x}_u \neq \mathbf{x}_o$ should be to drag $\mathbf{q}(\mathbf{x}_o)$ towards $\mathbf{p}^*(\mathbf{x}_o)$. This is the "unknown force" we observed earlier.

The other force affecting $\mathbf{q}(\mathbf{x}_o)$ during training is, of course, the update based on $\mathbf{x}_o$, driving $\mathbf{q}(\mathbf{x}_o)$ towards $\mathbf{e}_{y_o}$. How do these two forces balance over the course training? Early on, $\|\mathbf{e}_{y_u} - \mathbf{q}^t(\mathbf{x}_u)\|_2$ is relatively large for nearly any $\mathbf{x}_u$, because near initialization $\mathbf{q}^t(\mathbf{x}_u)$ will be relatively flat. Hence the size of the updates for "similar" $\mathbf{x}_u$ and the update for $\mathbf{x}_o$ should be comparable, meaning that, if there are at least a few "similar" training points, $\mathbf{q}^t(\mathbf{x}_o)$ will move towards $\mathbf{p}^*(\mathbf{x}_o)$. Throughout training, some of these similar samples will become well-classified, so that $\|\mathbf{e}_{y_u} - \mathbf{q}^t(\mathbf{x}_u)\|_2$ becomes small, and their updates will no longer exert much force on $\mathbf{q}(\mathbf{x}_o)$. Thus, the $\mathbf{x}_o$ updates begin to dominate, causing the zig-zag pattern as the learning path turns towards $\mathbf{e}_{y_o}$.

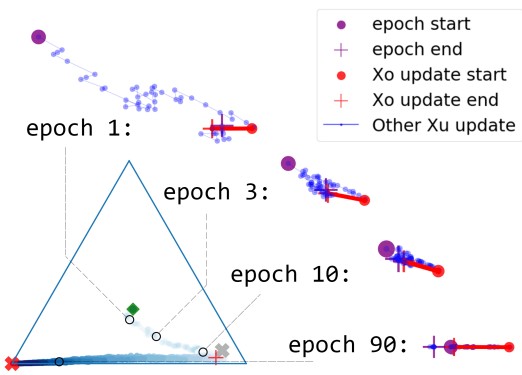

Figure 4: Updates of $\mathbf{q}(\mathbf{x}_o)$ over training.

For easy samples, where $\mathbf{p}^*$ and $\mathbf{e}_{y_o}$ are in the same direction, these forces agree and lead to fast convergence. On samples like the "medium" point in Figure 3, the two forces broadly agree early on, but the learning path deviates slightly towards $\mathbf{p}^*$ en route to $\mathbf{e}_{y_o}$.

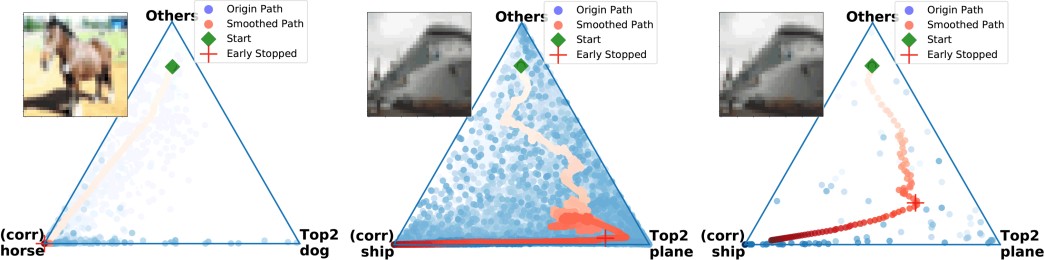

Figure 5: Learning path on CIFAR10. In the first two panels we record $\mathbf{q}^t$ for each batch, while in the last panel we record it for each epoch. See Appendix E for the learning paths of more samples.

Liu et al. (2020) prove that a similar pattern occurs while training a particular linear model on data with some mislabeled points, and specifically that stopping training at an appropriate early point will give better predictions than continuing training.

## 4  LEARNING PATHS ON REAL TASKS

Our analysis in Sections 2 and 3 demonstrate that the model can spontaneously refine the bad labels during one-hot training: the zig-zag learning path first moves towards the unknown $\mathbf{p}^*$. But is this also true on real data, for more complex network architectures? We will now show that they do, although seeing the patterns requires a little more subtlety.

We visualize the learning path of data points while training a ResNet18 (He et al., 2016) on CIFAR10 for 200 epochs as an example. The first panel of Figure 5 shows the learning path of an easy sample.[7] We can see that $\mathbf{q}^t$ converges quickly towards the left corner, the one-hot distribution for the correct class, because the color of the scattered points in that figure are quite light. At the early stopping epoch, $\mathbf{q}^t$ has already converged to $\mathbf{e}_y$. However, for a hard sample, it is very difficult to observe any patterns from the raw path (blue points): there are points almost everywhere in this plot. This is likely caused by the more complex network and dataset, as well as a high learning rate in early training. To find the hidden pattern in this high-variance path, we treat $\mathbf{q}^t$ as a time series signal with many high-frequency components. Thus we can collect its low-frequency components via a low-pass filter and then draw that, i.e., taking a exponential moving average (EMA) on $\mathbf{q}^t$ (red points). This makes it clear that, overall, the pattern zig-zags, first moving towards the unknown true label before eventually turning to memorize the wrong label.

This filtering method not only helps us observe the zig-zag pattern, but can also refine the labels. We use the following two experiments to verify this. First, we train the model on CIFAR10H using one-hot labels. As CIFAR10H provides $\mathbf{p}_{\text{hum}}$, which can be considered as an approximation of $\mathbf{p}^*$, we track the mean distance between $\mathbf{q}$ and $\mathbf{p}_{\text{hum}}$ during training. In the first panel of Figure 6, which averages the distance of all 10k training samples, the difference between the blue $\mathbf{q}_{\text{kd}}$ curve and red $\mathbf{q}_{\text{filt}}$ curve is quite small. However, we can still observe that the red curve is lower than the blue one before overfitting: filtering can indeed refine the labels. This trend is more significant when considering the most difficulty samples, as in the second panel: the gap between curves is larger.

To further verify the label refinement ability of filtering, we randomly choose 1,000 of the 50,000 CIFAR10 training samples, and flip their labels to a different $y' \neq y$. The last panel in Figure 6 tracks how often the most likely prediction of the network, $\operatorname{argmax} \mathbf{q}^t$, recovers the true original label, rather than the provided random label, for the corrupted data points. At the beginning of the model's training, the initialized model randomly guesses labels, getting about 10% of the 1,000 flipped labels correct. During training, the model spontaneously corrects some predictions, as the learning path first moves towards $\mathbf{p}^*$. Eventually, though, the model memorizes all its training labels. Training with the predictions from the 400th epoch corresponds to standard KD (no corrected points). Early stopping as in ESKD would choose a point with around 70% of labels corrected. The filtered path, though, performs best, with over 80% corrected.

---

[7]Without knowing $\mathbf{p}^*$, we instead use zig-zag score – see Appendix E – to measure the difficulty.

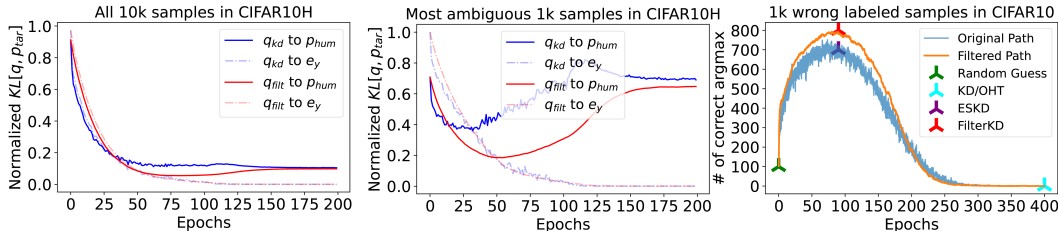

Figure 6: Filtering can refine the labels in both clean and noisy label case.

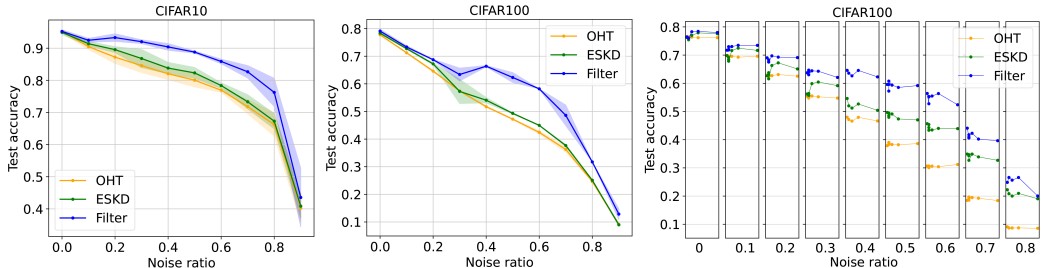

Figure 7: Test accuracy under different noise ratio $\sigma$. Solid lines are the means while shade region are the standard errors for 3 runs with different random seeds (shaded range is the standard error). Last panel compare the influence of different temperatures. Each thin rectangle plot represents a different $\sigma = \{0, 0.1, ..., 0.8\}$, in which we plot the results with different $\tau = \{0.5, 1, 2, 4, 10\}$.

## 5 FILTERING AS A METHOD FOR KNOWLEDGE DISTILLATION

Figure 6 gives a clear motivation for a new knowledge distillation method, which we call Filter-KD (Algorithm 1): train the student from the smoothed predictions of the teacher network. Specifically, we maintain a look-up table $\mathbf{q}_{\mathrm{smooth}} \in \mathbb{R}^{N \times K}$ to store a moving average of $\mathbf{q}_t$ for each training sample. Note that in one epoch, each $\mathbf{q}_{\mathrm{smooth}}(\mathbf{x}_n)$ will be updated only once. We check the early stopping criterion with the help of a validation set. Afterwards, the teaching supervision $\mathbf{q}_{\mathrm{smooth}}$ is ready, and we can train a student network under its supervision. This corresponds to using a moving average of the teacher model "in function space," i.e. averaging the outputs of the function over time.

Compared to ESKD, Filter-KD can avoid the extremely high variance of $\mathbf{q}^t$ during training. Unlike Figure 5, which plots $\mathbf{q}^t$ after every iteration of training, most practical implementations only consider early-stopping at the end of each epoch. This is equivalent to down-sampling the noisy learning path (as in the last panel of Figure 5), further exacerbating the variance of $\mathbf{q}^t$. Thus ESKD will likely select a bad $\mathbf{p}_{\mathrm{tar}}$ for many data points. Filter-KD, by contrast, has much more stable predictions.

We will show the effectiveness of this algorithm shortly, but first we discuss its limitations. Compared to ESKD, the running time of Filter-KD might be slightly increased. Furthermore, compared to the teaching model in ESKD, the Filter-KD requires a teaching table $\mathbf{q}_{\mathrm{smooth}}$, which will require substantial memory when the dataset is large. One alternative avoiding the need for this table would be to instead take an average "in parameter space," like e.g. momentum parameter updating as in Szegedy et al. (2015). We empirically find that, although this helps the model converge faster, it does not lead to a better teacher network; see Appendix I. Thus, although Filter-KD has clear drawbacks, we hope that our explanations here may lead to better practical algorithms in the future.

Similarly inspired by this spontaneous label refining mechanism (or early stopping regularization), Liu et al. (2020) and Huang et al. (2020) each propose algorithms aiming at the noisy label problem. We discuss the relationship between these three methods in Appendix H.

### 5.1 QUANTITATIVE RESULTS OF FILTER-KD

We now compare the performance of Filter-KD and other baselines on a real dataset. We focus on self-distillation and a fixed temperature $\tau = 1$ (except Table 2 and last panel in Figure 7), as we

| Noise $\sigma$ | 0% | 5% | 10% | 20% |
|---|---|---|---|---|
| **OHT** | 56.95 | 53.02 | 52.02 | 30.52 |
| **ESKD** | 58.61 | 53.53 | 52.99 | 36.55 |
| **FilterKD** | 59.32 | 56.43 | 55.51 | 40.81 |

Table 1: Results on TinyImageNet dataset.

| | Eff→Res | Res→Mob | Res→VGG |
|---|---|---|---|
| **Teacher** | 86.83 | 77.23 | 77.98 |
| **Student** | 78.09 | 72.58 | 71.04 |
| **ESKD** | 81.16 | 73.13 | 72.64 |
| **FilterKD** | 83.03 | 75.79 | 74.49 |

Table 2: Teacher→student, on CIFAR100.

want the only difference among these methods to be $\mathbf{p}_{tar}$. Thus we can conclude the improvement we achieved comes purely from the refined supervision. See Appendix D for other temperatures.

| | Accuracy | | | | ECE | | | |
|---|---|---|---|---|---|---|---|---|
| | **OHT** | **KD** | **ESKD** | **FilterKD** | **OHT** | **KD** | **ESKD** | **FilterKD** |
| CIFAR10 | 95.34 | 95.39 | 95.42 | 95.63 | 0.026 | 0.027 | 0.027 | 0.007 |
| CIFAR100 | 78.07 | 78.40 | 78.83 | 80.09 | 0.053 | 0.061 | 0.067 | 0.029 |

Table 3: Quantitative comparison of generalization performance for ResNet18 self-distillation (mean value of 5 runs). Refer to Table 6 for detailed results.

---

**Input:** Dataset $\{(\mathbf{x}_n, y_n)\}_{n=1}^{N}$, where $I_n = \{1, 2, ..., N\}$ is the index for each pair
*# Train the teacher*
Initialize a network model, initialize an $N \times K$ matrix called $\mathbf{q}_{smooth}$
Go through the entire dataset, calculate $\mathbf{q}_{smooth}[n] = \text{Softmax}(\text{model}(\mathbf{x}_n))$
**for** $epoch = 1, 2, ..., T$ **do**
    **for** $n \in \{1, 2, ..., N\}$ in random order **do**
        $\hat{p} = \text{Softmax}(\text{model}(\mathbf{x}_n))$
        $\mathbf{q}_{smooth}[n] = (1 - \alpha) \cdot \mathbf{q}_{smooth}[n] + \alpha \cdot \hat{p}$
        Update parameters based on $loss = \text{CrossEntropy}(\hat{p}, y_n)$,
    **end for**
    Check the early stopping criterion; stop training if satisfied
**end for**
*# Train the student*
For each input $\mathbf{x}_n$, set $\mathbf{p}_{tar} = \mathbf{q}_{smooth}[n]$; train the network under the supervision of $\mathbf{p}_{tar}$

**Algorithm 1:** Filter-KD. $\alpha$ controls the cut-off frequency of low-pass filter (0.05 here).

---

The first task we consider is noisy-label classification, where we train and validate the model on a dataset with some labels randomly flipped. After training, all the models are evaluated on a clean held-out test set. The experiments are conducted on CIFAR (Figure 7) and TinyImageNet (Table 1), under different noise ratios $\sigma$: $\sigma = 0.1$ means 10% of the labels are flipped. In Figure 7, an interesting trend can be observed: the enhancement brought by Filter-KD is not significant when $\sigma$ is too small or too large. That is reasonable because for small $\sigma$, few samples have bad labels, thus the possible enhancement might not be large. When $\sigma$ is too high, the labels of similar points become less reliable, and the learning path will no longer head as reliably towards $\mathbf{p}^*$. Thus, for very high noise ratios, the performance of Filter-KD decays back towards that of OHT.

Filter-KD also outperforms other methods in both accuracy and calibration on the clean dataset, as illustrated in Table 3. This remains true in the more common KD case when the teacher network is bigger than the student; see results in Table 2. Note that in Table 2, we no longer keep $\tau = 1$, as the baselines are more sensitive to $\tau$ when the teacher is large (see Appendix D for more discussion); instead we optimize $\tau$ for each setting. Here "Eff" is short for EfficientNet-B1 (Tan & Le, 2019), "Res" is short for ResNet18, "Mobi" is short for MobileNetV2 (Sandler et al., 2018), "VGG" is short for VGG-11 (Simonyan & Zisserman, 2015).

In summary, the explanation in Section 4 and experimental results in this section demonstrate that better supervisory signal, which can be obtained by Filter-KD, can enhance the prediction performance in both clean and noisy-label case. (It is also possible that the algorithm improves perfor-

mance for other reasons as well, especially in the clean label case; the influence of temperature may be particularly relevant.)

## 6   RELATED WORK

**Human label refinement**   Peterson et al. (2019) use a distribution of labels obtained from multiple annotators to replace the one-hot label in training, and find that doing so enhances both the generalization performance of trained models and their robustness under adversarial attacks. However, this approach is relatively expensive, as it requires more annotation effort than typical dataset creation techniques; several experienced annotators are required to get a good estimate of $\mathbf{p}^*$.

**Label smoothing**   Another popular method is label smoothing (Szegedy et al., 2016), discussed in the previous sections. We believe that suitable smoothing can decrease $\|\mathbf{p}_{\text{tar}} - \mathbf{p}^*\|_2$ to some extent, but the lower bound of this gap should be large, because label smoothing treats each class and each sample in the same way. Much prior research (e.g. Müller et al., 2019) shows that KD usually outperforms label smoothing as well.

**KD and ESKD**   Knowledge distillation, by contrast, can provide a different $\mathbf{p}_{\text{tar}}$ for each training sample. KD was first proposed by Hinton et al. (2015) for model compression. Later, Furlanello et al. (2018); Cho & Hariharan (2019); Zhang et al. (2019); Yuan et al. (2020) found that distillation can help even in "self-distillation," when the teacher and student have identical structure. Our work applies to both self-distillation and larger-teacher cases.

Another active direction in research about KD is finding explanations for its success, which is often still considered somewhat mysterious. Tang et al. (2020) decompose the "dark knowledge" provided by the teacher into three parts: uniform, intra-class and inter-class. This explanation also overlaps with ours: if we observe the samples with different difficulty in the same class, we are discussing intra-class knowledge; if we observe samples with two semantically similar classes, we then have inter-class knowledge. Among previous work, the most relevant study is that of Menon et al. (2021). Our Hypothesis 1 is suggested by their main claim, which focuses on the question of *what is a good* $\mathbf{p}_{tar}$. Our work builds off of theirs by helping explain *why this good* $\mathbf{p}_{tar}$ *emerges, and how to find a better one*. Besides, by looking deeper into the learning path of samples with different difficulty, our work might shed more light on KD, anti-noisy learning, supervised learning, overfitting, etc.

**Noisy label task**   Learning useful information from noisy labels is a long-standing task (Angluin & Laird, 1988; Natarajan et al., 2013). There are various ways to cope with it; for instance, Menon et al. (2019) use gradient clipping, Patrini et al. (2017) use loss correction, Huang et al. (2020) change the supervision during training, and Zhang et al. (2020) employ extra information. It is possible to combine KD-based methods with traditional ones to further enhance performance; our perspective of learning paths also provides further insight into why KD can help in this setting.

## 7   DISCUSSION

In this paper, we first claim that better supervision signal, i.e., $\mathbf{p}_{\text{tar}}$ that is closer to $\mathbf{p}^*$, leads to better generalization performance; this is supported by results of Menon et al. (2021) and further empirical suggestions given here. Based on this hypothesis, we explain why LS and KD outperform one-hot training using experiments on a toy Gaussian dataset. To further understand how such better supervision emerges, we directly observe the behavior of samples with different difficulty by projecting them on a 2-D plane. The observed zig-zag pattern is well-explained by considering the tradeoff between two forces, one pushing the prediction to be near that of similar inputs, the other pushing the prediction towards its training label; we give an intuitive account based on Proposition 1 of why this leads to the observed zig-zag pattern.

To apply these findings on real tasks, in which the data and network are more complex, we conduct low-pass filter on the learning path, and propose Filter-KD to further enhance $\mathbf{p}_{\text{tar}}$. Experimental results on various settings (datasets, network structures, and label noise) not only show the advantage of the proposed method as a method for a teacher in knowledge distillation methods, but also help verify our analysis of how generalization and supervision work in training neural network classifiers.

ACKNOWLEDGEMENTS

This research was enabled in part by support provided by the Canada CIFAR AI Chairs program, WestGrid, and Compute Canada.

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

## A   MISCELLANEOUS BACKGROUND

**Calculation of ECE**   Expected calibration error is a measurement about how well the predicted confidence represent the true correctness likelihood (Guo et al., 2017). For example, if our model gives 100 predictions, each with confidence, say, $q(y = k|\mathbf{x}) \in [0.7, 0.8)$. Then we might expect there are $70 \sim 80$ correct predictions among those 100 ones. To calculate this, we first uniformly divide $[0, 1]$ into $M$ bins, with each bin represented by $I_m \in \left(\frac{m-1}{M}, \frac{m}{M}\right]$ and $B_m$ is all the $\mathbf{x}$ samples whose confidence falls into $I_m$. Then, ECE is calculated as:

$$\text{ECE} = \sum_{m=1}^{M} \frac{|B_m|}{n} \left| acc(B_m) - conf(B_m) \right|, \tag{5}$$

where $acc(B_m) = \frac{1}{|B_m|} \sum_{i \in B_m} \mathbb{1}(\hat{y}_i = k_i)$, $conf(B_m) = \frac{1}{|B_m|} \sum_{i \in B_m} q(y = k_i|\mathbf{x}_i)$, $k_i$ is the true label and $\hat{y}_i = argmax[q(y|\mathbf{x}_i)]$ is the model's prediction. All the ECE mentioned in this paper is calculated by setting $M = 10$.

**Barycentric coordinate system**   When visualizing the learning path, it is problematic to directly choose two dimensions and draw them in the Cartesian coordinate system, as illustrated in the first panel in Figure 8. In geometry, a suitable way to project a 3-simplex vector onto a 2-D plane is converting it to a point in a barycentric coordinate system. Specifically, we have three basis vectors: $\mathbf{v}_0 = [0, 0]^\mathsf{T}$, $\mathbf{v}_1 = [1, 0]^\mathsf{T}$, and $\mathbf{v}_2 = \left[\frac{1}{2}, \frac{\sqrt{3}}{2}\right]^\mathsf{T}$, which are the corner and two edges of an equilateral triangle respectively. Then the 2D-coordinate is calculated as $(x, y) = [\mathbf{v}_0; \mathbf{v}_1; \mathbf{v}_2] \cdot \mathbf{q}$. So every points in the left corner of a Cartesian system plane can be converted to the triangle in a barycentric system, as illustrated in the last panel in Figure 8.

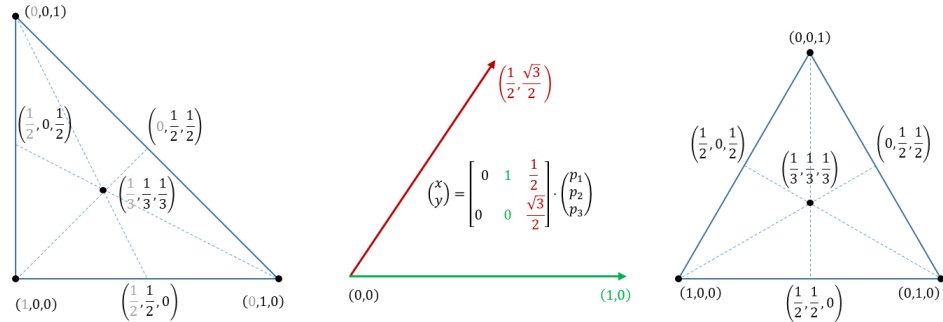

Figure 8: How to project a probability vector on to the plane of Barycentric coordinate system.

## B   RISK ESTIMATE VARIANCE BOUND

The following result is a generalization of Proposition 3 of Menon et al. (2021), whose proof we replicate and extend here:

**Proposition 2.** *Let $L$ be any bounded loss, $L(y, \hat{y}) \leq \ell < \infty$, and consider $R_{tar}$ of (3). For any predictor $f : \mathcal{X} \to \Delta^K$, we have that*

$$\mathbb{E}_{\mathcal{D}'} \left[ (R_{tar}(f, \mathcal{D}') - R(f))^2 \right] \leq \frac{1}{N} \underset{\mathbf{x}}{\mathrm{Var}} \left[ \mathbf{p}_{tar}^\mathsf{T} \mathbf{L}(f(\mathbf{x})) \right] + \xi,$$

*where $\xi$ can be any of the following seven quantities:*

$$\ell^2 K \left( \mathbb{E}_{\mathbf{x}} \|\mathbf{p}_{tar}(\mathbf{x}) - \mathbf{p}^*(\mathbf{x})\|_2 \right)^2$$

$$\ell^2 \left( \mathbb{E}_{\mathbf{x}} \|\mathbf{p}_{tar}(\mathbf{x}) - \mathbf{p}^*(\mathbf{x})\|_1 \right)^2$$

$$2\ell^2 \left( \mathbb{E}_{\mathbf{x}} \sqrt{\mathrm{KL}(\mathbf{p}_{tar}(\mathbf{x}) \,\|\, \mathbf{p}^*(\mathbf{x}))} \right)^2 \qquad 2\ell^2 \mathbb{E}_{\mathbf{x}} \mathrm{KL}(\mathbf{p}_{tar}(\mathbf{x}) \,\|\, \mathbf{p}^*(\mathbf{x}))$$

$$2\ell^2 \left( \mathbb{E}_{\mathbf{x}} \sqrt{\mathrm{KL}(\mathbf{p}^*(\mathbf{x}) \,\|\, \mathbf{p}_{tar}(\mathbf{x}))} \right)^2 \qquad 2\ell^2 \mathbb{E}_{\mathbf{x}} \mathrm{KL}(\mathbf{p}_{tar}(\mathbf{x}) \,\|\, \mathbf{p}^*(\mathbf{x}))$$

$$\ell^2 \mathbb{E}_{\mathbf{x}} \left[ \mathrm{KL}(\mathbf{p}_{tar}(\mathbf{x}) \,\|\, \mathbf{p}^*(\mathbf{x})) + \mathrm{KL}(\mathbf{p}^*(\mathbf{x}) \,\|\, \mathbf{p}_{tar}(\mathbf{x})) \right].$$

*Proof.* To begin,

$$\mathbb{E}_{\mathcal{D}'} \left[ (R_{\mathrm{tar}}(f, \mathcal{D}') - R(f))^2 \right] = \mathrm{Var}_{\mathcal{D}'} [R_{\mathrm{tar}}(f, \mathcal{D}') - R(f)] + \left( \mathbb{E}_{\mathcal{D}'} [R_{\mathrm{tar}}(f, \mathcal{D}') - R(f)] \right)^2.$$

For the variance term, since $R(f)$ is a constant and $R_{\mathrm{tar}}$ is an average of $N$ i.i.d. terms, we get

$$\mathrm{Var}_{\mathcal{D}'} [R_{\mathrm{tar}}(f, \mathcal{D}') - R(f)] = \frac{1}{N} \mathrm{Var}_{\mathbf{x}} [\mathbf{p}_{\mathrm{tar}}(\mathbf{x})^\top \mathbf{L}(f(\mathbf{x}))].$$

The other term, as $R_{\mathrm{tar}}$ is an average of i.i.d. terms and $R(f) = \mathbb{E}_{\mathbf{x}} \mathbf{p}^*(\mathbf{x})^\top \mathbf{L}(f(\mathbf{x}))$, is

$$\left( \mathbb{E}_{\mathcal{D}'} [R_{\mathrm{tar}}(f, \mathcal{D}') - R(f)] \right)^2 = \left( \mathbb{E}_{\mathbf{x}} (\mathbf{p}_{\mathrm{tar}}(\mathbf{x}) - \mathbf{p}^*(\mathbf{x}))^\top \mathbf{L}(f(\mathbf{x})) \right)^2.$$

For the first bound, we apply the Cauchy-Schwarz inequality,

$$(\mathbf{p}_{\mathrm{tar}}(\mathbf{x}) - \mathbf{p}^*(\mathbf{x}))^\top \mathbf{L}(f(\mathbf{x})) \leq \|\mathbf{p}_{\mathrm{tar}}(\mathbf{x}) - \mathbf{p}^*(\mathbf{x})\|_2 \|\mathbf{L}(f(\mathbf{x}))\|_2;$$

since the elements of $\mathbf{L}(f(\mathbf{x}))$ are each at most $\ell$, the term $\|\mathbf{L}(f(\mathbf{x}))\|_2$ is at most $\sqrt{K}\ell$.

For the other bounds, we instead apply Hölder's inequality, yielding

$$(\mathbf{p}_{\mathrm{tar}}(\mathbf{x}) - \mathbf{p}^*(\mathbf{x}))^\top \mathbf{L}(f(\mathbf{x})) \leq \|\mathbf{p}_{\mathrm{tar}}(\mathbf{x}) - \mathbf{p}^*(\mathbf{x})\|_1 \|\mathbf{L}(f(\mathbf{x}))\|_\infty \leq \ell \|\mathbf{p}_{\mathrm{tar}}(\mathbf{x}) - \mathbf{p}^*(\mathbf{x})\|_1.$$

The KL bounds follow by Pinsker's inequality and then Jensen's inequality. The last bound, for the Jeffreys divergence, combines the two KL bounds. $\square$

## C   TOY GAUSSIAN DATASET

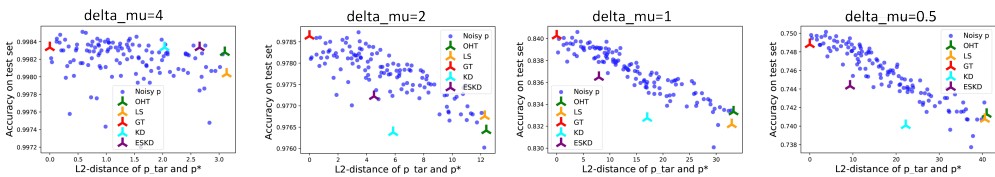

Figure 9: Correlation between test accuracy and $\|\mathbf{p}_{\mathrm{tar}} - \mathbf{p}^*\|_2$ for different settings.

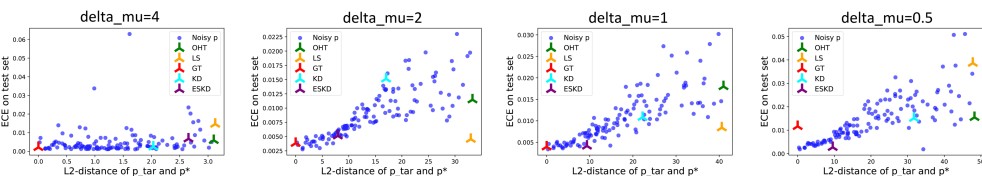

Figure 10: Correlation between test ECE and $\|\mathbf{p}_{\mathrm{tar}} - \mathbf{p}^*\|_2$ for different settings.

In Sections 2 and 3, we apply a toy Gaussian dataset to verify two facts derived from Hypothesis 1, so as the learning path of specific samples. Here we provide more details about this dataset.

| | Run1 | | | | Run2 | | | | Run3 | | | |
|---|---|---|---|---|---|---|---|---|---|---|---|---|
| | $\tau$=0.5 | $\tau$=1 | $\tau$=2 | $\tau$=10 | $\tau$=0.5 | $\tau$=1 | $\tau$=2 | $\tau$=10 | $\tau$=0.5 | $\tau$=1 | $\tau$=2 | $\tau$=10 |
| **OHT** | 94.87 | - | - | - | 95.15 | - | - | - | 94.48 | - | - | - |
| **ESKD** | 95.02 | 95.27 | **95.34** | 95.05 | 95.41 | **95.41** | 95.39 | 94.76 | 94.11 | 94.65 | **94.88** | 94.75 |
| **FilterKD** | 95.18 | **95.87** | 95.66 | 94.81 | 95.11 | 95.56 | **95.71** | 94.81 | 95.09 | **95.31** | 95.22 | 95.16 |
| **OHT** | 77.85 | - | - | - | 78.23 | - | - | - | 77.99 | - | - | - |
| **ESKD** | 78.28 | 78.56 | **79.28** | 79.09 | 78.50 | 78.20 | **79.3** | 78.81 | 78.12 | 78.31 | **78.84** | 78.36 |
| **FilterKD** | 78.43 | 79.57 | **79.72** | 79.65 | 79.23 | **79.61** | 79.45 | 79.01 | 79.75 | 80.32 | **80.41** | 79.99 |

Table 4: The influence of temperature in self-distillation on CIFAR10/100 dataset.

**Generate the dataset**    Here, we have $N$ samples, each sample a 3-tuple $(\mathbf{x}, y, \mathbf{p}^*)$. To get one sample, we first select the label $y = k$ following an uniform distribution over all $K$ classes. After that, we sample the input signal $\mathbf{x}|_{y=k} \sim \mathcal{N}(\boldsymbol{\mu}_k, \sigma^2 I)$, where $\sigma$ is the noisy level for all the samples. $\boldsymbol{\mu}_k$ is the mean vector for all the samples in class $k$. Each $\boldsymbol{\mu}_k$ is a 30-dim vector, in which each dimension is randomly selected from $\{-\delta_\mu, 0, \delta_\mu\}$. Such a process is similar to selecting 30 different features for each class. Finally, we calculate the true Bayesian probability of this sample, i.e., $p^*(y|\mathbf{x})$.

**Calculate the ground truth probability**    We use the fact that $p^*(y|\mathbf{x}) \propto p(\mathbf{x}|y)p(y)$. As $y$ follows an uniform distribution, we have $p^*(y|\mathbf{x}) = \frac{p(\mathbf{x}|y=k)}{\sum_{j \neq k} p(\mathbf{x}|y=j)}$. Following $p(\mathbf{x}|y = k) \sim \mathcal{N}(\boldsymbol{\mu}_k, \sigma^2 I)$, we find $p^*(y|\mathbf{x})$ should have a Softmax form, i.e., $p = \frac{e^{s_k}}{\sum_{j \neq k} e^{s_j}}$. Specifically, we have:

$$p^*(y = k|\mathbf{x}) = \frac{e^{s_k}}{\sum_{j \neq k} e^{s_j}}; \quad s_i = -\frac{1}{2\sigma^2} \|\mathbf{x} - \boldsymbol{\mu}_i\|_2^2. \tag{6}$$

**Setup of experiments in Figure 1**    in this experiment, we generate a toy Gaussian dataset with $K = 3$, $\sigma = 2$ and $N = 10^5$. To reduce the variance of test error, we make a train/valid/test split with ratio [0.05 0.05, 0.9]. We train an MLP with 3 hidden layers, each with 128 hidden units and ReLU activations. We first conduct experiments on some baseline settings, i.e., learning from one-hot supervision (OHT for short), from smoothed label supervision (LS for short), from a converged teacher's prediction (KD for short), and from an early-stopped teacher's prediction (ESKD for short). In OHT case, an NN is trained under the supervision of $\mathbf{e}_y$. If we train this NN until the convergence of training accuracy, we obtain the $\mathbf{p}_{\text{tar}}$ for the KD case. If we select the snapshot of that NN based on the best validation accuracy, we obtain the $\mathbf{p}_{\text{tar}}$ for the ESKD case. If we directly set $\mathbf{p}_{\text{tar}} = 0.9\mathbf{e}_y + 0.1\mathbf{u}$, where $\mathbf{u}$ is an uniform distribution over $K$ classes, we obtain the supervision for LS case. With different supervisions, we train a new network with identical structure until the validation accuracy no longer increase. Furthermore, to see a trend between generalization ability and $\|\mathbf{p}_{\text{tar}} - \mathbf{p}^*\|_2$, we run the experiment 200 times under different noisy supervisions.

## D    TEMPERATURE IN DIFFERENT KD METHODS

Temperature is an important hyper-parameter in different kinds of KD methods, which might influence the performance a lot. In this appendix, we will discuss why we prefer $\tau = 1$.

**The role played by $\tau$**    In general, the loss function in KD has the form:

$$L = \beta \cdot \left(\frac{1}{\tau^2}\right) \cdot \mathcal{H}(\mathbf{q}^\tau, \mathbf{p}_{\text{tar}}^\tau) + (1 - \beta) \cdot \mathcal{H}(\mathbf{q}, \mathbf{e}_y), \tag{7}$$

where $\beta \in [0, 1]$ is another hyper-parameter to trade-off the importance between one-hot label and teacher's predictions. Furthermore, for this loss, the gradient of $L$ to logits $\mathbf{z}$ is:

$$\frac{\partial L}{\partial \mathbf{z}_i} = \beta \cdot \left(\frac{1}{\tau}\right) \cdot (\mathbf{q}_i^\tau - \mathbf{p}_{\text{tar}_i}^\tau) + (1 - \beta) \cdot (\mathbf{q}_i - \mathbf{e}_{y_i}). \tag{8}$$

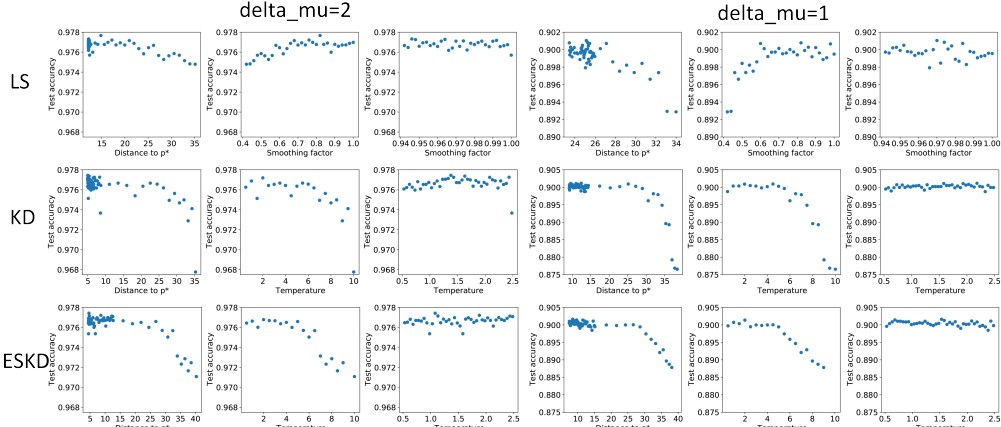

Figure 11: The influence of label smoothing factor on LS (first row) and different $\tau$ on KD (other rows) under two different settings of the toy dataset. It is clear that these hyper-parameters won't influence the performance too much as long as they are in a good region.

**Why we use $\tau = 1$**  The most important reason for us to choose $\tau = 1$ (as well as $\beta = 1$), even with the risk of providing sub-optimal performance, is that we want to observe how much enhancement is brought by refining the label. In our mind, KD's success comes from the following two aspects:

  (a)  better label (i.e., $\mathbf{p}_{\text{tar}}$ is better than $\mathbf{e}_y$);
  (b)  better learning dynamics (soften $\mathbf{q}$ to $\mathbf{q}^\tau$ make the training easier).

The first one provides the student with more useful knowledge, and the second one helps the student to extract it. The focus of our paper is the improvement in labels. When the temperature is not 1, both (a) and (b) are influenced, as we are using $\mathbf{q}^\tau$ (i.e., the smoothed student output) to match $\mathbf{p}_{\text{tar}}^\tau$ (i.e., the smoothed target) during training and using $\mathbf{q}$ to inference during testing. If we fix $\tau = 1$, the only difference between Filter-KD, ESKD, KD, LS, and OHT training is $\mathbf{p}_{\text{tar}}$. Under such a condition, we believe the fact that Filter-KD/ESKD outperforms OHT (strictly better in each run) is enough to conclude KD methods can provide better labels. Furthermore, as Filter-KD is proposed to mitigate the high-variance issue in ESKD, and we can directly observe the zig-zag learning path of samples with bad labels, we believe it is reasonable to make the conclusion in the paper.

Furthermore, using $\tau \neq 1$ doesn't substantially change our analysis. As illustrated in Equation (8), the gradient shows that $\mathbf{q}^\tau$ moves towards $\mathbf{p}_{\text{tar}}^\tau$ in each update. Hence our analysis in Proposition 1 still holds, which means the trade-off between the two forces still exists.

$\tau = 1$ **is reasonably good in our settings** In fact, the optimal $\tau$ depends on many settings, e.g., teacher's size, optimizer, learning rate (scheduler), and etc: there is still no consensus on what is the best choice of $\tau$ for all settings. So for the experiments in the main context, we fix $\tau$ and fine-tune other hyper-parameters to achieve good results. However, to verify this choice of $\tau$ is not terrible, we performed a coarse grid search on both CIFAR and toy examples. As illustrated in Table 4 and Fig-

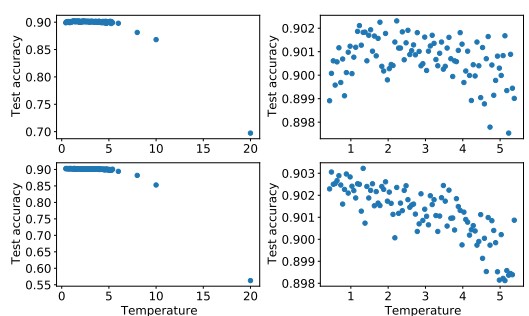

Figure 12: First row: large teacher to small student; second row: self-distill. Left column: $\tau$ in a larger range; right column: $\tau$ in a small range.

ure 11, choosing $\tau \in [1, 2]$ seems to be good choice: the performance doesn't significantly decay until $\tau$ is too large.

Some readers might also curious about why our results don't match the common notion that the optimal $\tau$ should be 4, which is first provided in Hinton et al. (2015) and followed by much later

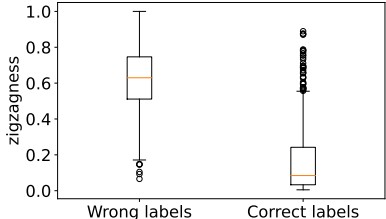

Figure 13: The learning path of samples with correct and wrong labels.

work (e.g. Gao et al., 2018; Zagoruyko & Komodakis, 2017; Cho & Hariharan, 2019). Some works also use a grid search to conclude that a temperature as high as 20 should be the best choice (Yuan et al., 2020; Kim et al., 2019). We find this mismatch comes from the relative difference between the network size of teacher and student. In short, when distilling from a large teacher to a small student (as most of the cases in aforementioned works), high $\tau$ is preferred. When conducting self-distillation, $\tau$ need not be that high. For example, Zhang et al. (2019) and Roth et al. (2021) claim $\tau = 1$ is the best choice, while Zhang & Sabuncu (2020) claim $\tau \approx 2$ is the best. To further verify this, we compare the temperature trend between two cases on the toy dataset. In Figure 12, the first row is distilling a 10-layer, 256-width MLP to a 3-layer, 32-width MLP; the second row is self-distillation between 3-layer, 32-width MLPs.

At the same time, we also run a grid search the smoothing factor $\alpha$ of our Filter-KD in Table 5.

| Smoothing $\alpha$ | 0.01 | 0.05 | 0.1 | 0.2 | 0.5 | 1 (ESKD) |
|---|---|---|---|---|---|---|
| **FilterKD** | 79.50 | **80.00** | 79.83 | 79.59 | 78.48 | 78.39 |

Table 5: The influence of the smoothing factor in FilterKD on CIFAR100 (mean of 3 runs). For comparison, OHT obtained 77.64.

## E   HOW REPRESENTATIVE IS THE ZIG-ZAG PATTERN?

In the main paper, we only visualize the zig-zag learning path of a few samples in Figures 3 and 5. The readers might then wonder whether this pattern is representative across the whole dataset. To verify this, we define a quantitative metric called **zig-zag score**. Specifically, we first calculate the integral of each dimension of the prediction:

$$Q_{i \in \{1,...,K\}} \triangleq \sum_{t=1}^{T} \mathbf{q}_i^t(\mathbf{x}_n). \tag{9}$$

We then use the highest $Q_i$ among $i \in \{1, ..., K\} \setminus \{y\}$, where $y$ is the label's class, as the zig-zag score. In other words, we focus on the behavior of $\mathbf{q}$ on those dimensions that are *not* the training label. If this score is large, we might expect the neighbouring samples exert high influence on the path, and vice versa. Note that as we have $\sum_i Q_i = $ constant for any sample (as $\mathbf{q}^t$ is a K-simplex), this zig-zag score might correlated with C-score of Jiang et al. (2021). However, their focuses are different: C-score is similar to $-Q_{i=y}$ and focuses more on how fast the prediction converge to the label; zig-zag score, on the other hand, is more about how much the path deviates towards a different class, possibly the label close to $\mathbf{p}^*$.

With this score, we test the correlation between base difficulty and zig-zag score on toy datasets under different settings. From Figure 14, it is safe to conclude that samples with higher base difficulty will have a more zig-zagging path during training. We also compare the expected zig-zag score of the 1000 samples with flipped label in CIFAR10 (recall the experiment in Figure 6). In Figure 13, it is clear that the zig-zagness of these wrong label samples (which we are sure have high base difficulty) is significantly higher than the average.

Finally, we also randomly select some data samples in each class of CIFAR10 and visualize their learning paths. Figure 15 shows random samples for the noisy label experiment. It is clear that the

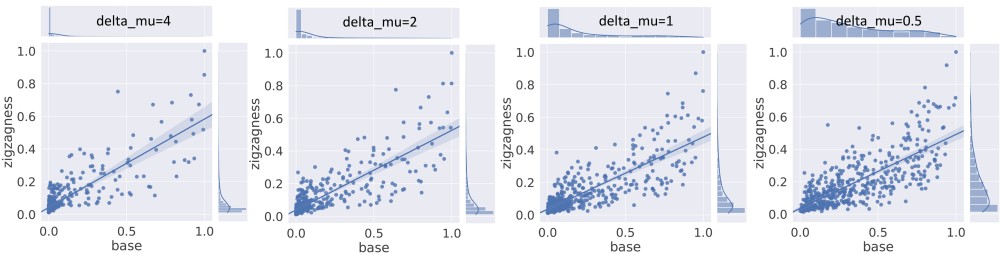

Figure 14: Correlation between base difficulty and zig-zag score in four different toy datasets.

learning path of easy samples with correct labels converge fast while that of samples with wrong labels is zig-zag. In Figure 16, which shows the learning path with high zig-zag score when the training set is clean, we can still observe some samples with zig-zag path. These samples might be ambiguous, with a quite flat $\mathbf{p}^*$. However, as we do not know the true $\mathbf{p}^*$ of these samples, it is impossible to provide a result like Figure 13.

## F    DISTANCE GAP UNDER DIFFERENT SUPERVISIONS

Figure 2 provides the distance gap between $\mathbf{q}$ and corresponding $\mathbf{p}^*$ for each training sample in different training stages, under the supervision of one-hot label $\mathbf{e}_y$. From the results in this figure, we notice that the behavior of hard samples contributes more to the success of ESKD. Here we further visualize how these gaps changes when the model is trained under different types of supervision, i.e., ground truth, LS and ESKD.

In the first row, which is the result of label smoothing, we see the $\|\mathbf{p}_{\mathrm{tar}} - \mathbf{p}^*\|_2$ (i.e., red dots) has a "V"-shape. The vertex location depends on the smoothing parameter we choose. However, as discussed previously, label smoothing has only one parameter to control $\mathbf{p}_{\mathrm{tar}}$, which is like using a linear model to fit a high-order function. So, although a proper smoothing value can bring better supervision than one-hot label, its upper bound might be limited. Regarding the training dynamics, we can see a similar trend as the results shown in the one-hot case, i.e., all the samples first move down and then converge to $\mathbf{p}_{\mathrm{tar}}$. From the middle panel, we might expect label smoothing to outperform the one-hot case, because the scatters are closer to the x-axis, which represents the ground truth $\mathbf{p}^*$.

The second and the third rows demonstrate the KD and ESKD case. Results in the KD case are quite similar to the one-hot case in Figure 2, because the converged $\mathbf{p}_{\mathrm{tar}}$ is close to $\mathbf{e}_y$. However, we can still expect KD to outperform one-hot training, because the $\mathbf{p}_{\mathrm{tar}}$ is closer to $\mathbf{p}^*$ than $\mathbf{e}_y$ is. The last panel in this row also demonstrates that $\|\mathbf{q} - \mathbf{p}^*\|_2$ might be smaller than $\|\mathbf{p}_{\mathrm{tar}} - \mathbf{p}^*\|_2$, which can be considered as an explanation for why iterated self-distillation, e.g., Born Again Networks (Furlanello et al., 2018), can improve the performance. For the ESKD case, we see the overfitting almost disappear: the distribution of the blue points do not change too much after the early stopping criterion is satisfied.

In the last row, which illustrates the training under the supervision of $\mathbf{p}^*$, we find all the blue points move toward the x-axis, i.e., their $\mathbf{p}_{\mathrm{tar}} = \mathbf{p}^*$, and finally converge to it. There is also no overfitting in this case. From the last two panels, we see the disperse of blue points is significantly smaller than all other settings, which means the network's prediction is quite close to $\mathbf{p}^*$. Hence the performance of this case is the best.

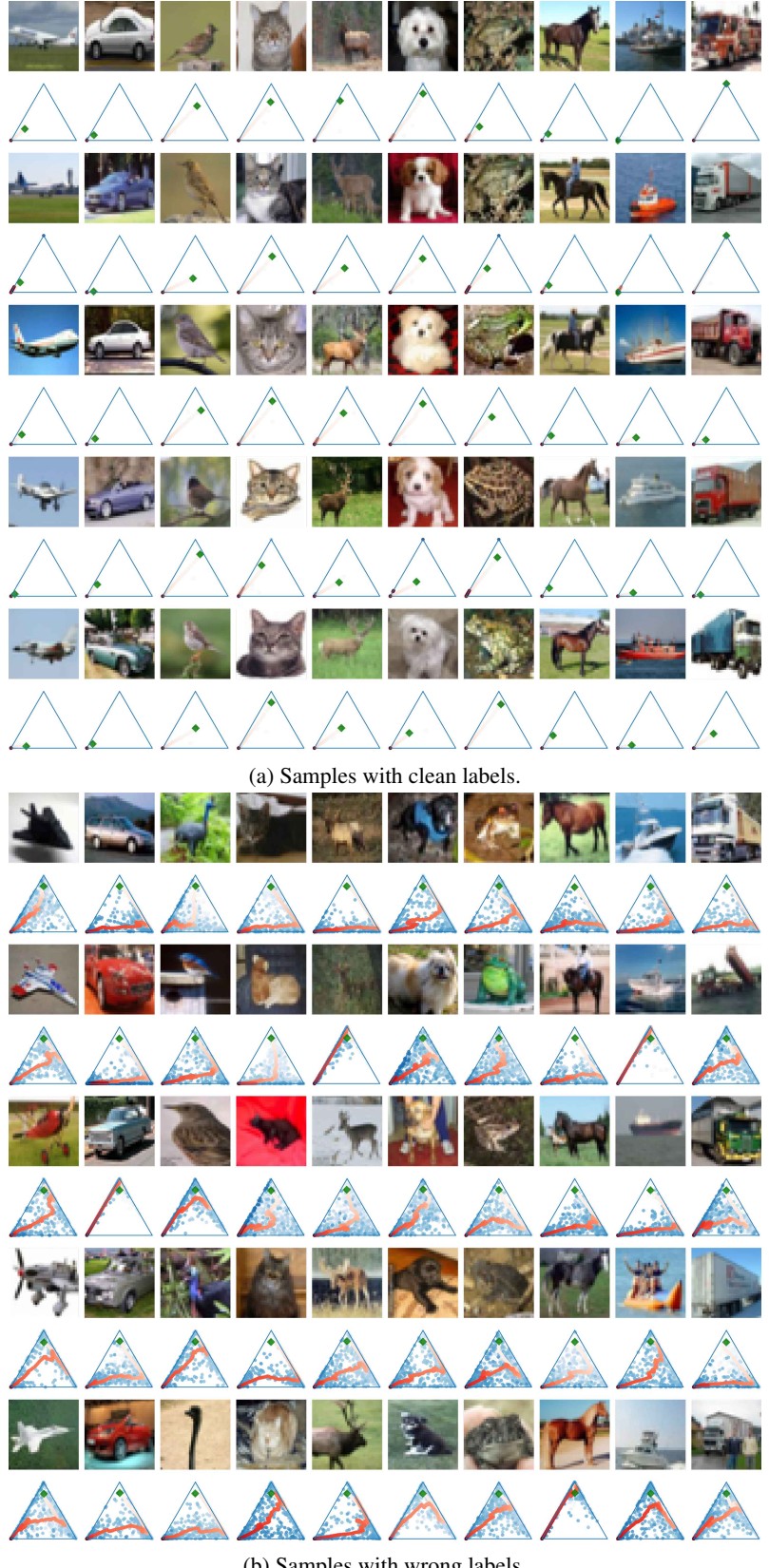

(a) Samples with clean labels.

(b) Samples with wrong labels.

Figure 15: Random selection of samples in CIFAR10 with 1000 flipped labels.

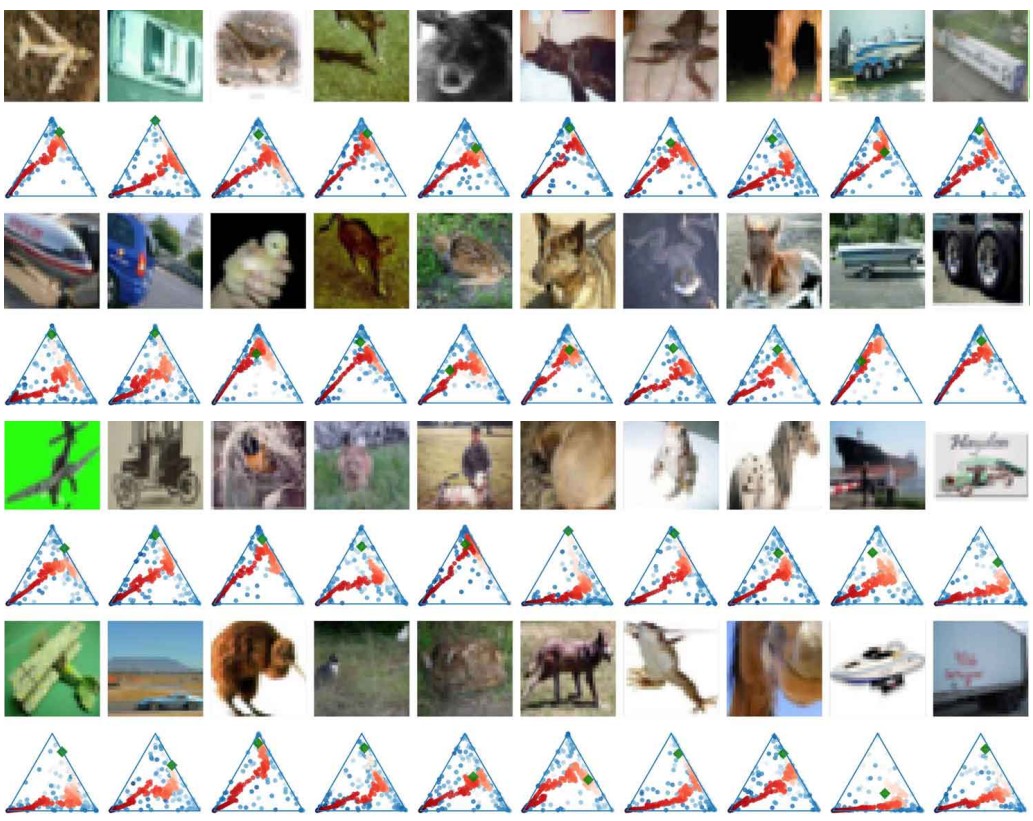

Figure 16: Random selection of samples with high zig-zag score in clean CIFAR10.

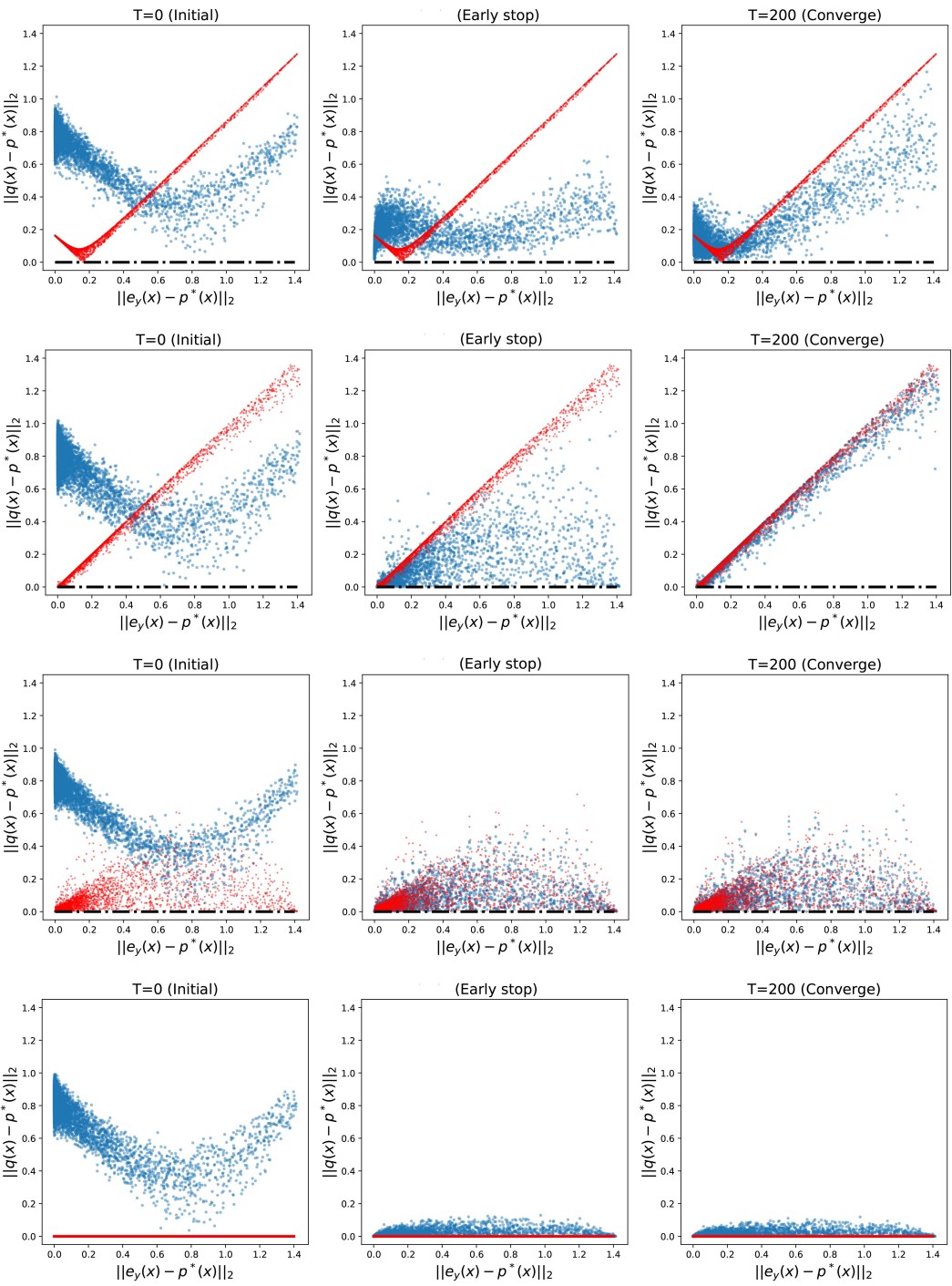

Figure 17: Distance gap of each sample under different supervision: label smoothing, KD, ESKD, and ground-truth training.

## G  MORE ABOUT THE DECOMPOSITION AND NTK MODEL

*Proof of Proposition 1.* Recall $\mathbf{z}(\mathbf{x}) = f(\mathbf{w}, \mathbf{x})$ is the vector of output logits, and $\mathbf{q} = \mathrm{Softmax}(\mathbf{z})$ is the output probability. We are taking a step of SGD observing $\mathbf{x}_u$, and observing the change in predictions on $\mathbf{x}_o$. We begin with a Taylor expansion,

$$\underbrace{\mathbf{q}^{t+1}(\mathbf{x}_o)}_{K \times 1} - \underbrace{\mathbf{q}^t(\mathbf{x}_o)}_{K \times 1} = \underbrace{\nabla_\mathbf{w} \mathbf{q}^t(\mathbf{x}_o)|_{\mathbf{w}^t}}_{K \times d} \cdot \underbrace{(\mathbf{w}^{t+1} - \mathbf{w}^t)}_{d \times 1} + \mathcal{O}\big(\|\mathbf{w}^{t+1} - \mathbf{w}^t\|^2\big).$$

To evaluate the leading term, we plug in the definition of SGD and repeatedly use the chain rule:

$$\underbrace{\nabla_\mathbf{w} \mathbf{q}^t(\mathbf{x}_o)|_{\mathbf{w}^t}}_{K \times d} \cdot \underbrace{(\mathbf{w}^{t+1} - \mathbf{w}^t)}_{d \times 1} = \big( \underbrace{\nabla_\mathbf{z} \mathbf{q}^t(\mathbf{x}_o)|_{\mathbf{z}^t}}_{K \times K} \cdot \underbrace{\nabla_\mathbf{w} \mathbf{z}^t(\mathbf{x}_o)|_{\mathbf{w}^t}}_{K \times d} \big) \cdot \big( -\eta \underbrace{\nabla_\mathbf{w} L(\mathbf{x}_u)|_{\mathbf{w}^t}}_{1 \times d} \big)^\mathsf{T}$$

$$= \underbrace{\nabla_\mathbf{z} \mathbf{q}^t(\mathbf{x}_o)|_{\mathbf{z}^t}}_{K \times K} \cdot \underbrace{\nabla_\mathbf{w} \mathbf{z}^t(\mathbf{x}_o)|_{\mathbf{w}^t}}_{K \times d} \cdot \big( \underbrace{-\eta \nabla_\mathbf{z} L(\mathbf{x}_u)|_{\mathbf{z}^t}}_{1 \times K} \cdot \underbrace{\nabla_\mathbf{w} \mathbf{z}^t(\mathbf{x}_u)|_{\mathbf{w}^t}}_{K \times d} \big)^\mathsf{T}$$

$$= -\eta \underbrace{\nabla_\mathbf{z} \mathbf{q}^t(\mathbf{x}_o)|_{\mathbf{z}^t}}_{K \times K} \cdot \big[ \underbrace{\nabla_\mathbf{w} \mathbf{z}^t(\mathbf{x}_o)|_{\mathbf{w}^t}}_{K \times d} \cdot \underbrace{\big( \nabla_\mathbf{w} \mathbf{z}^t(\mathbf{x}_u)|_{\mathbf{w}^t} \big)^\mathsf{T}}_{d \times K} \big] \cdot \underbrace{\big( \nabla_\mathbf{z} L(\mathbf{x}_u)|_{\mathbf{z}^t} \big)^\mathsf{T}}_{K \times 1}$$

$$= \eta \cdot \mathcal{A}^t(\mathbf{x}_o) \cdot \mathcal{K}^t(\mathbf{x}_o, \mathbf{x}_u) \cdot \big( \mathbf{p}_{\mathrm{tar}}(\mathbf{x}_u) - \mathbf{q}^t(\mathbf{x}_u) \big).$$

For the higher-order term, using as above that

$$\mathbf{w}^{t+1} - \mathbf{w}^t = -\eta \nabla_\mathbf{w} \mathbf{z}^t(\mathbf{x}_u)|_{\mathbf{w}^t}^\mathsf{T} \cdot \big( \mathbf{p}_{\mathrm{tar}}(\mathbf{x}_u) - \mathbf{q}^t(\mathbf{x}_u) \big)$$

and noting that, since the vectors are probabilities, $\|\mathbf{p}_{\mathrm{tar}}(\mathbf{x}_u) - \mathbf{q}^t(\mathbf{x}_u)\|$ is bounded, we have that

$$\mathcal{O}\big( \|\mathbf{w}^{t+1} - \mathbf{w}^t\|^2 \big) = \mathcal{O}\big( \eta^2 \|(\nabla_\mathbf{w} \mathbf{z}(\mathbf{x}_u)|_{\mathbf{w}^t})^\mathsf{T}\|_{\mathrm{op}}^2 \|\mathbf{p}_{\mathrm{tar}}(\mathbf{x}_u) - \mathbf{q}^t(\mathbf{x}_u)\|^2 \big) = \mathcal{O}\big( \eta^2 \|\nabla_\mathbf{w} \mathbf{z}(\mathbf{x}_u)\|_{\mathrm{op}}^2 \big). \quad \square$$

In the decomposition,

$$\mathcal{A}^t(\mathbf{x}_o) = \nabla_z \mathbf{q}^t(\mathbf{x}_o)|_{\mathbf{z}^t} = \begin{bmatrix} q_1(1 - q_1) & -q_1 q_2 & \cdots & -q_1 q_K \\ -q_2 q_1 & q_2(1 - q_2) & \cdots & -q_2 q_K \\ \vdots & \vdots & \ddots & \vdots \\ -q_K q_1 & -q_K q_2 & \cdots & q_K(1 - q_K) \end{bmatrix}, \tag{10}$$

which is a symmetric positive semi-definite (PSD) matrix[8] with trace $\mathrm{tr}(\mathcal{A}^t(\mathbf{x}_o)) = 1 - \sum_{i=1}^K q_i^2$. As we have $\sum_i q_i = 1$, it is easy to check the trace of this matrix is larger at the beginning of training (when $\mathbf{q}$ tends to be flat) than that at the end of training ($\mathbf{q}$ tends to be peaky), as illustrated by most panels in Figure 18. Given that the trace of a matrix is the sum of its eigenvalues and $\mathcal{A}^t(\mathbf{x}_o)$ is PSD, smaller $\mathrm{tr}(\mathcal{A}^t(\mathbf{x}_o)) = 1 - \sum_{i=1}^K q_i^2$ means this matrix will tend to shrink its inputs more. Hence the change in predictions tends to decrease when $\mathbf{q}^t$ becomes more peaky.

The second term in that expression, $\mathcal{K}^t(\mathbf{x}_o, \mathbf{x}_u)$, is the outer product of gradients at $\mathbf{x}_o$ and $\mathbf{x}_u$. Intuitively, if their gradients have similar directions, this matrix is large, and vice versa. This matrix is known as the empirical neural tangent kernel, and it can change through the course of training as the network's notion of "similarity" evolves. For appropriately initialized very wide networks trained with very small learning rates, $\mathcal{K}^t$ remains almost constant during the course of training, and is almost independent of the initialization of the network parameters; the kernel it converges to is known as the neural tangent kernel (Jacot et al., 2018; Arora et al., 2019).

**Learning Dynamics of $\mathbf{q}(\mathbf{x}_o)$**   In Proposition 1, we decompose $\mathbf{q}^{t+1}(\mathbf{x}_o) - \mathbf{q}^t(\mathbf{x}_o)$ into three parts; we use this to analyze what the learning path of a training sample might be like in Section 3.3. Here we will provide more detailed illustration of the two groups of force imposed on $\mathbf{q}(\mathbf{x}_o)$.

Specifically, $\mathbf{q}^{t+1}(\mathbf{x}_o) - \mathbf{q}^t(\mathbf{x}_o)$ will be influenced by two variables, i.e., updating sample $\mathbf{x}_u$ and time $t$. We discuss their effects separately. For $\mathbf{x}_u$, only the last two terms, i.e., $\mathcal{K}^t(\mathbf{x}_o, \mathbf{x}_u)$ and $(\mathbf{p}_{\mathrm{tar}}(\mathbf{x}_u) - \mathbf{q}^t(\mathbf{x}_u))$ depends on it.

---

[8]The matrix $\mathcal{A}$ can be observed to be the covariance matrix of a categorical distribution with item probabilities $q$, and hence PSD.

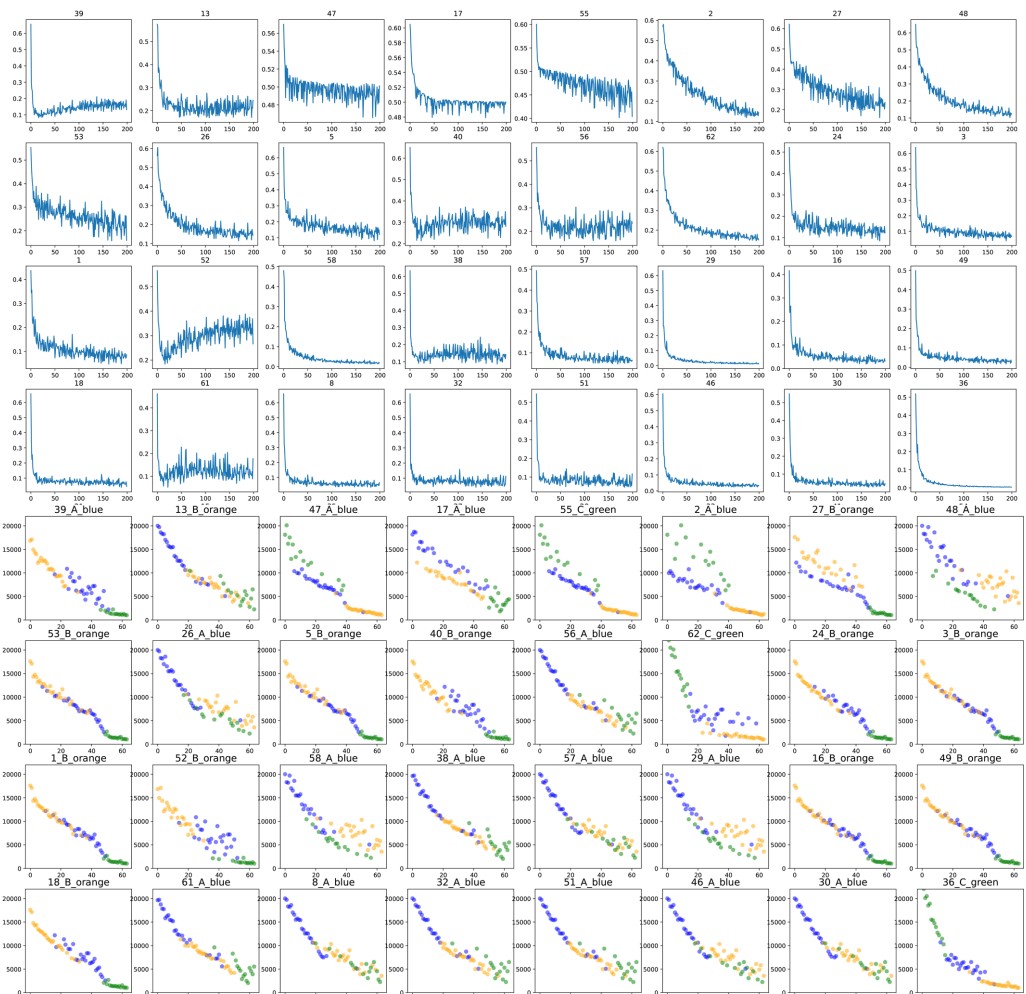

Figure 18: Upper panels: how $\mathrm{tr}(\mathcal{A}^t(\mathbf{x}_n))$ changes during training. Each panel represents a specific $\mathbf{x}_n$. The panels are ordered by their integral difficulty, from left-to-right and up-to-down. The x-axis is the number of epochs, and the y-axis is $\mathrm{tr}(\mathcal{A}^t(\mathbf{x}_n))$. Lower panels: the correlation between $\cos(\mathbf{x}_o, \mathbf{x}_u)$ and $\mathrm{tr}(\mathcal{K}^0)$. Panels are ordered by the integral difficulty of $\mathbf{x}_o$. The subtitle of the panel gives the sample ID, the class it belongs to (i.e., A, B, or C) and the color of their corresponding class (i.e., A is blue, B is orange and C is green).

In Section 3.3, we claim that if $\mathbf{x}_o$ and $\mathbf{x}_u$ are similar, the norm of $\mathcal{K}^0(\mathbf{x}_o, \mathbf{x}_u)$ might be large, and vice versa. Here we empirically demonstrate this using a toy Gaussian dataset, as illustrated in Figure 18. The figure shows how the similarity between $\mathbf{x}_o$ and $\mathbf{x}_u$ correlates with $\mathrm{tr}(\mathcal{K}^0(\mathbf{x}_o, \mathbf{x}_u))$. Each panel represents one $\mathbf{x}_o$, which is claimed in the title of the subfigures. The x-axis represents the rank of the cosine similarity between observed $\mathbf{x}_o$ and all $N$ training samples (including itself). The y-axis is the trace of $\mathcal{K}^0(\mathbf{x}_o, \mathbf{x}_u)$. The color of each scatter point is the true label of $\mathbf{x}_u$. From the figure, we can observe a clear decreasing trend, which means smaller similarity leads to larger $\mathrm{tr}(\mathcal{K}^0(\mathbf{x}_o, \mathbf{x}_u))$. Additionally, the term $(\mathbf{p}_{\mathrm{tar}}(\mathbf{x}_u) - \mathbf{q}^t(\mathbf{x}_u))$ provides a direction that $\mathbf{q}^t(\mathbf{x}_o)$ should move towards.

We claim in Section 3.3 that at any point in the input space, the labels of these input samples might follow the ground truth distribution, i.e., $p^*(y|\mathbf{x})$. Hence most of the neighbouring $\mathbf{x}$ might pull $\mathbf{q}(\mathbf{x}_o)$ towards its ground truth $p^*(y|\mathbf{x}_o)$. We will discuss the norm of this term when discussing the influence of $t$. In short, at any time, the neighboring $\mathbf{x}_u$ will impose stronger force on $\mathbf{x}_o$ and the direction of the force roughly points to the ground truth $p^*(y|\mathbf{x}_o)$.

As discussed, $\mathcal{K}^t$ is roughly constant over $t$ in the very-wide limit. For finite width networks, however, it adapts to reflect the network's new "understanding" of similarities. For instance, it might learn that certain types of images are more semantically similar than the randomly-initialized network thought. This does not fundamentally change our intuitions as long as it doesn't happen too often, but could potentially lead to more complicated zig-zag patterns as the network's estimate of $\mathbf{p}^*$ from neighboring points perhaps improves over time.

Over the course of training, $\mathcal{A}^t(\mathbf{x}_o)$ and $(\mathbf{p}_{\text{tar}}(\mathbf{x}_u) - \mathbf{q}^t(\mathbf{x}_u))$ will also change. In practical regimes, none of these terms have an easy analytical expression w.r.t. $t$: $\mathbf{q}^t$ is quite complicated. Thus, we provide some intuition, with experimental verification. In Figure 18, we show how $\text{tr}(\mathcal{A}^t(\mathbf{x}_o))$ depends on $\mathbf{q}^t$: flat $\mathbf{q}^t$ leads to larger trace. A similar trend also exists in the norm of $(\mathbf{p}_{\text{tar}}(\mathbf{x}_u) - \mathbf{q}^t(\mathbf{x}_u))$. As the initialized $\mathbf{q}$ tend to be flat, updates of any samples will influence network's parameters a lot. When the training progresses, those easy samples converge fast, so their $\|\mathbf{p}_{\text{tar}}(\mathbf{x}_u) - \mathbf{q}^t(\mathbf{x}_u)\|_2$ and $\text{tr}(\mathcal{A}^t)$ become small. However, as the $\mathbf{q}^t$ for the hard sample is still far away from its $\mathbf{e}_y$, the large $\|\mathbf{p}_{\text{tar}}(\mathbf{x}_u) - \mathbf{q}^t(\mathbf{x}_u)\|_2$ and $\text{tr}(\mathcal{A}^t)$ will finally drag $\mathbf{q}^t$ back towards the one-hot distribution, as illustrated in Figure 3.

## H  COMPARISON TO LIU ET AL. (2020) AND HUANG ET AL. (2020)

The main claim of this paper is that better supervisory signals can enhance the generalization ability of the trained model. Inspired by the success of KD, we find that the neural network can spontaneously refine those "bad" labels during training by observing their learning path. The learning path of those hard samples will first move towards their true $p(y|x)$ and then converge to their label $p_{\text{tar}}$ or $e_y$. We explain why this phenomenon occurs by expanding the gradients of each training sample. This phenomenon is also explained in Proposition 1, and formally proved for a particular softmax regression model by Liu et al. (2020). As a complement, we propose an explanation using an NTK model, and experimentally verify it by observing the learning path and distance gap during training.

Another difference between these two works is that we consider the problem of refining supervisory signals, while Liu et al. (2020) consider correcting wrong labels (a special case of "bad" supervision). Our work provides additional emphasis and empirical study for the clean-label case.

Regarding the algorithm, Liu et al. (2020) design an effective regularization term inspired by early stopping regularization. They apply exponential moving average (EMA) on the model's output when calculating this regularization term to further enhance the performance. This method is similar to that proposed by Huang et al. (2020), who switch between optimizing the training loss and an objective based on the EMA of the model over the course of training.

Although it bears significant similarity to these methods, Filter-KD does not change the course of training the teacher model. Rather, we propose (based on the high variance of the zig-zag learning paths) to simply use the the EMA of that model's outputs as a teacher for later distillation.

We suspect that these three algorithms work because of essentially the same underlying principle, whether we think of this as being based on the zig-zag learning path or as early-stopping regularization. We expect that this principle will be helpful moving forward in the field's understanding of the learning dynamics of SGD methods for neural networks.

## I  LOW PASS FILTER ON NETWORK PARAMETERS

In Section 4, we point out the high variance issue of the traditional KD methods after observing the learning path of those hard samples. We then propose a Filter-KD algorithm to smooth the output of each training sample during training. Such a low pass filtering method is quite similar to the momentum mechanism used in self-supervised learning, e.g., from the classic method of Ruppert (1988) to MOCO (He et al., 2020), which conduct low pass filtering on each parameter of the network. As mentioned before, the proposed Filter-KD algorithm might require more memory when the dataset becomes larger, because $\mathbf{q}_{\text{smooth}}$ would record every training sample's prediction.

Conducting low pass filtering on network parameters might be a good way to solve this. To verify whether this method works, we train a ResNet18 on CIFAR100, using one-hot supervision. At the beginning of training, we copy the training model and call it tracking model. At the end of each

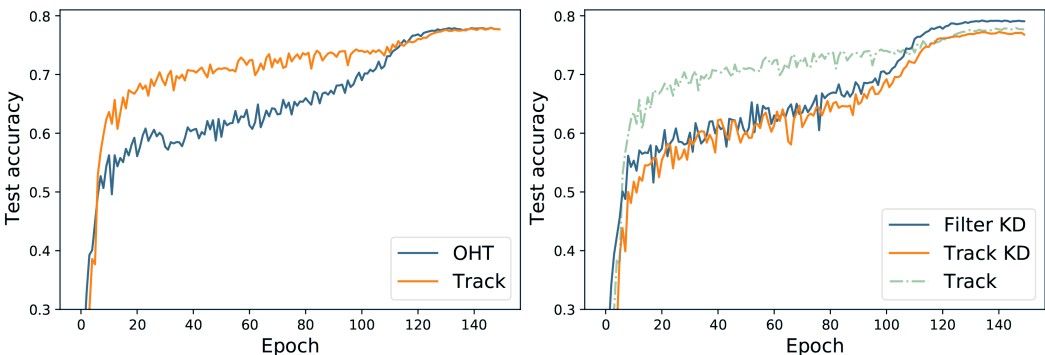

Figure 19: Behavior of the tracking model (with low pass filter on each parameters). ResNet18 trained for 150 epochs on CIFAR100.

update (i.e., each batch), the parameters of training model is updated as usual while the tracking model's parameters are updated with momentum. Specifically, for the training model, $\mathbf{w}_{\text{train}}^{t+1} = \mathbf{w}_{\text{train}}^{t} + \eta \nabla L$, for the tracking model, $\mathbf{w}_{\text{track}}^{t+1} = (1 - \alpha)\mathbf{w}_{\text{track}}^{t} + \alpha\mathbf{w}_{\text{train}}^{t+1}$. We train the model for 150 epochs, and show the learning curve of test accuracy in the left panel of Figure 19. We see the performance of the tracking model converges faster than the training model, which is reasonable because filtering parameters is regularizing the training process. However, the converging accuracy of this two models are the same. At the same time, we find this tracking model is not as good as $\mathbf{q}_{\text{smooth}}$ when teaching a new model, as illustrated in the right panel of Figure 19.

As this paper mainly focuses on the behavior of the model's prediction, the discussion and experiments of filtering in parameter space is limited. However, as many papers demonstrates the effectiveness of this momentum mechanism, we think it is important to explore the relationship further.

| | Test ACC | | | | | Test ECE | | | | |
|---|---|---|---|---|---|---|---|---|---|---|
| | Run1 | Run2 | Run3 | Run4 | Run5 | Run1 | Run2 | Run3 | Run4 | Run5 |
| **OHT** | 95.35 | 95.30 | 95.42 | 95.23 | 95.42 | 0.027 | 0.027 | 0.026 | 0.025 | 0.025 |
| **KD** | 95.30 | 95.38 | 95.42 | 95.44 | 95.42 | 0.027 | 0.027 | 0.027 | 0.026 | 0.026 |
| **ESKD** | 95.29 | 95.41 | 95.39 | 95.58 | 95.42 | 0.026 | 0.029 | 0.025 | 0.028 | 0.027 |
| **FilterKD** | 95.66 | 95.68 | 95.49 | 95.76 | 95.58 | 0.005 | 0.006 | 0.008 | 0.011 | 0.006 |
| **OHT** | 78.27 | 78.31 | 77.97 | 77.78 | 78.02 | 0.053 | 0.053 | 0.052 | 0.053 | 0.054 |
| **KD** | 78.64 | 78.55 | 78.03 | 78.18 | 78.49 | 0.060 | 0.057 | 0.062 | 0.066 | 0.059 |
| **ESKD** | 78.84 | 78.73 | 78.85 | 78.97 | 78.74 | 0.065 | 0.066 | 0.067 | 0.070 | 0.066 |
| **FilterKD** | 79.87 | 79.93 | 80.19 | 80.22 | 80.23 | 0.028 | 0.026 | 0.034 | 0.028 | 0.031 |

Table 6: Each run of results in Table 3. In each run, the initialization of student networks for different methods are controlled to be the same.

