# OpenReview forum: "Better Supervisory Signals by Observing Learning Paths"
_ICLR.cc/2022/Conference — ICLR 2022 Poster_

### Official Review · Reviewer_bjwd · 2021-10-29

**Correctness:** 3
**Technical Novelty And Significance:** 3
**Empirical Novelty And Significance:** 3
**Recommendation:** 8
**Confidence:** 4

**Main Review:**

**This review has been updated since it was originally posted. See thread below.**

## Strengths:
- The paper is, for the most part, well-structured and easy to read. Argument and ideas are developed step-by-step, formulated and substantiated formally, based on previous work.
- Establishing hypothesis up front.
- Starting from a toy example, building up the contribution from initial observations, and finally moving on to more complex/realistic scenarios after.
- Tackles a relevant problems for the ML community, namely knowledge distillation and noisy labels.

## Weaknesses:
- The concept of noise is not well-defined, to the point where the problem definition becomes confusing: is the noise big enough to flip the target label? More precisely, does `arg max_i p*(y|x)_i = arg max_i e_yi` always hold? If it does, the hard sample from Figure 3 does not seem to comply. If the equality doesn't hold, the problem definition is more about data with wrongly labeled samples; in this case, the insights from section 3 will be contingent on the number of samples for which the equality mentioned above doesn't hold (i.e., mislabeled samples). This aspect has not been addressed in the paper. Moreover, the concept of easy, medium and hard samples does not explicitly compensate for mislabeled samples, and hard samples *could* trivially correspond to the mislabeled samples.
- KD, ESKD and LS depend substantially on a temperature hyper-parameter which the paper does not address. Having the wrong temperature can drastically affect the performance of these baselines.
- The labels from ESKD can be seen as a special case of the proposed Filter-KD where alpha = 1. Therefore, one could expect the labels from an ESKD teacher to be similar to those of Filter-KD if the teacher were trained using less epochs (in other words, if the early-stopping criterion forces the teacher to stop sooner). Even though ESKD is a valid baseline, is it the most compelling one to justify the introduction of Filter-KD?
- Does not use a real dataset (i.e., not a toy dataset) for which `p*` is available like CIFAR10H **[1]** to evaluate Hypothesis 1.

### Detailed comments:
- The abstract is not a clear description of the paper, lacks context and coherence. It is not clear what the contributions are (is it a criterion to measure generalization, a function to measure difficulty or a novel way to do knowledge distillation).

- Increase text size in all figures (at least as big as the figure's caption).

> In this paper, we first clarify what makes supervision good for a classification problem, and then explain two existing label refining methods, label smoothing and knowledge distillation (KD), in terms of our proposed criterion.

*The paper clarifies or proposes a new criterion to quantify generalization performance?*

> we look deeper into the learning path of the network’s predicted distribution for training samples with different difficulty.

*What is the 'learning path'? How is difficulty defined? Make it clear that these are definitions that will be introduced later on.*

- Evaluate labels produced by teachers trained on KD, ESKD, LS and Filter-KD wrt. human annotated labels from CIFAR10H **[1]**.

- Proposition 3 from Menon et al.: Lacks context. I recommend explaining further where it comes from to make the argument more self-contained.

- Figure 1b-c: The range on the y-axis is rather small (coverage of 0.7% for accuracy and 0.02 on ECE), exaggerating correlation between accuracy and similarity between `p*` and `p_tar`. Also, KD and LS depend on a temperature parameter, which according to the supplementary material, has only been tested with with one setup.

> ...leads to better generalization performance, as measured either by accuracy (ACC) or expected calibration error (ECE)

*It is critical to define the term "generalization" here. If these are the two criteria that are going to define generalization, or are distribution shifts also being considered?*

- Figure 2: It would be interesting to see the [distribution of the entire dataset for each axis](https://matplotlib.org/stable/_images/sphx_glr_scatter_hist_001.png). The scatterplot can hide how dense the distribution is (and how thin the tails are).

- Figure 3 (hard sample): How can the `p*()` label for the hard sample be so close to the bottom right corner (say [0.1, 0.8, 0.1]), and yet the one-hot encoding be all the way across, on the bottom left corner ([1, 0, 0])? It looks as if the samples were mislabeled. Refer to the first point under 'Weaknesses'.

- Figure 3: How representative is this behavior across the dataset? How many easy/medium/hard samples are there?

- Figure 3: The paths on the right are close to impossible to see and interpret.

- Proposition 1: `x_o` has not been defined within the proposition.

> however, for a harder sample (middle), it is very difficult to observe any patterns

*Capitalization*

- Figure 4: What does the black dashed line mean?

- Section 5.1: What labels are used for the students' test set? Is it the original OHT or a label set by the corresponding teacher?

- Section 5 & Figure 5: How does Filter-KD compare to models trained using LS and KD (with optimum temperature)? Any reason to leave them out?

### References
[1] Peterson, Joshua C., et al. "Human uncertainty makes classification more robust." Proceedings of the IEEE/CVF International Conference on Computer Vision. 2019.

**Summary Of The Paper:**

This paper introduces a method for doing knowledge distillation with noisy labels. The contribution consists on representing distilled labels as a weighted moving average of the predicted labels from a teacher as it trains (using noisy one-hot encoded labels). Finally, a student is trained using the distilled labels.

**Summary Of The Review:**

The paper is easy to follow and has contributions that are sufficiently motivated. Empirical evaluation shows that the proposed method is more robust to noisy labels. However, the problem definition (therefore the scope of the paper) is not clear, and the core contribution Filter-KD has not been evaluated to an extent that justifies its simplicity.

**UPDATE: After discussing with the authors (see thread below) I have updated my review to reflect my view of the paper.**

---

> ### Author Response · Authors · 2021-11-23
> **We answer most of the concerns in new Appendices C, D, E. (Part I)**
>
> Thank you for identifying the strengths of this work. We appreciate the thorough review and have made several enhancements based on your comments. If you have other concerns in mind, we would be happy to take them into consideration.
>
> Please also note the [general response](https://openreview.net/forum?id=Iog0djAdbHj&noteId=4nBozEauGKg).
>
> > Weakness-->1. The concept of noise is not well-defined...
>
> First, it is worth noting that the stated equation may well not hold for points that are not "mislabelled." Consider points in a three-class problem with p* = (0.1, 0.8, 0.1). If the training data is sampled directly from this distribution, 20% of the data – which is correctly labeled by any reasonable definition – will not satisfy the equation above. Even in CIFAR10H, there are 79 samples whose label does not agree with the argmax.
>
> This highlights a very good point: the references to "noise," "difficulty," "easy/medium/hard," and so on in the previous version were indeed quite confusing. So in this version, we only use base-difficulty, the L2 distance between the label, i.e., $p_{tar}, e_y,$ and the ground truth $p(y|x)$, i.e., $p^*$, to quantify how hard the learning task is (or equivalently, how bad the labels are).
>
> For noise added to the label, as in Figure 1, we state in the paper that "whenever we mention adding noise to the label, we are adding a noise factor to each dimension of that label, and then normalize it to be a distribution. Hence a too large noise can flip the label." For experiments in Figures 6 and 7, we flip the one-hot label to $y'\neq y$, rather than adding continuous noise to the target.
>
> We also claim that the zig-zag pattern can be considered as a spontaneous label refining mechanism (one which FilterKD can help). Hence, when the task is harder, there might be more bad labels for us to refine, hence the improvement of FilterKD to OHT case can be larger. Considering the one-hot labels, there are two typical cases for a label to be bad: one is that $p^*$ is flat, which means the sample x is ambiguous; another is that $p^*$ is peaky and close to true y, but the label is flipped to $y'$, which makes $e_y'$ far away from $p^*$. In the toy dataset, we change the distance between the mean values of different Gaussian distributions to control the ratio of ambiguous samples. In the noisy-label CIFAR case, we flip some labels to control the ratio of wrong labels. Note that even in the clean-label CIFAR case, our FilterKD still consistently outperforms the OHT case, which means there exist some bad labels in the CIFAR training set.
>
> > 2. KD, ESKD, and LS depend substantially...
>
> Please refer to Q1 in the overall response. We also provide a thorough discussion about the influence of temperature in Appendix B.
>
> > 3. The labels from ESKD can...
>
> This claim might not be precise. Note that FilterKD is proposed to solve the high-variance problem of the learning path in a real dataset, especially when a large learning rate is applied at the beginning of training. Hence we cannot expect the labels from an ESKD teacher to be similar to those of a FilterKD teacher at the early stage of training, because the learning path of ESKD can be very unpredictable. See the last panel in Figure 5, in which the original path disperses almost the whole region, which means the ESKD's output in time T and T+1 can be very different. But after filtering, the path is smoother and the zig-zag pattern becomes clear.
>
> This process is quite similar to low-pass filtering in signal processing, which helps to recover the true information in a low signal-to-noise-ratio scenario. Thus we believe the introduction of FilterKD is justified after observing the original learning path. (On the toy dataset, the path is clean, so FilterKD is no better than ESKD.)
>
> Also see Figure 6, which demonstrates FilterKD outperforms ESKD for any choice of stopping epoch on CIFAR, and Table 5 (Appendix B), which shows how the choice of alpha influences performance.
>
> > 4. Does not use a real dataset...
>
> Please refer to Q2 in the overall response and Appendix E. But in short, 1. we do not believe labels in CIFAR10H is a perfect estimation of $p^*$, for which we have now added experimental support; 2. we believe using the toy dataset makes the paper easier to understand.
>
> ## Continue in Part II

---

> > ### Author Response · Authors · 2021-11-23
> > **We answer most of the concerns in Appendix B,C, D. (Part II)**
> >
> > > Detailed Comments --> 1. The abstract is not a clear description...
> >
> > We re-write the abstract part to highlight the flow of the paper. Please also refer to Q0 in the general response.
> >
> > > 2. Increase text size in all figures...
> >
> > Thanks for pointing this out. We fixed most of them in this version. But due to the time limits, some figures (like Figure 7) are still not in good shape; we'll fix them in the next version.
> >
> > > 3. The paper clarifies or proposes a new criterion...
> >
> > No, we just clarify that the L2-distance between a label and its ground truth $p^*$ can measure how good this label is. We also claim that training a model with a better label can make the model generalize well on a held-out test set. We don't advocate using some kind of estimate of $||p_\text{tar} - p^*||_2$ to replace validation accuracy as a measure of generalization or anything like that.
> >
> > > 4. Evaluate labels produced by teachers ...
> >
> > As mentioned previously, we don't think the evaluation on CIFAR10H makes a lot of sense, as we do not believe CIFAR10H provides a perfect estimation of $p^*$. If you nonetheless think it makes sense to explore further, we can do some further investigation in this area later (maybe during the discussion period).
> >
> > >5. Proposition 3 from Menon et al....
> >
> > In this version, we only mention the main results of Proposition 3 from Menon et al., i.e., equation (4). We added a slight clarification when we introduce the result. From this expression, we can conclude that a smaller L2 distance leads to better risk and hence better test performance.
> >
> > >6. Figure 1b-c: The range on the y-axis...
> >
> > Generally for the temperature, please refer to Q1 in the overall response and Appendix B, in which we provide some interesting analysis about the influence of temperature.
> >
> > For the y-axis part, as we mentioned before, the possible enhancement we can get by refining the labels depends on how many bad labels there are. Hence in Figures 8 and 9 (Appendix A), we see the range on the y-axis becomes larger when the task is harder. Anyway, even for the results in Figure 1, a Spearman rank correlation test shows the correlation is strong: the correlation coefficient and p-value are $-0.93$ and $1.9\times10^{-53}$ for Figure 1b, and $0.895$ and $2.7\times10^{-43}$ for Figure 1c.
> >
> > >7. It is critical to define the term "generalization"...
> >
> > In terms of generalization, we consider only the standard notion of generalizing to a test set from the same distribution; we now clarify this in footnote 1.
> >
> > Out-of-distribution generalization is a far more complex topic; we are not aware of any particular relationship to knowledge distillation.
> >
> > > 8. Figure 2: It would be interesting...
> >
> > Thanks for this good suggestion. We add the distribution in Figures 2 and 13. Comparing the histogram in the second and the last panel in Figure 2, we see the overfitted network indeed increases the ratio of the bad predictions.
> >
> > >9. Figure 3 (hard sample)...
> >
> > Please see the previous discussion in response to weakness 1.
> >
> > >10. Figure 3: How representative...
> >
> > Thanks for pointing out this. We provide a thorough analysis about this problem in Q2 in general response, so as in Appendix C. It is clear that zig-zagness is indeed highly correlated with base difficulty. In other words, the NN has the ability of refining bad labels and FilterKD can assist this process.
> >
> > >11. Figure 3: The paths on the right...
> >
> > Sorry, we agree that these figures were indeed hard to understand. We remade the figure and added some explanation in Figure 4. We use these visualizations to give readers an intuition about how the output probability changes during training in specific epochs. One can treat them as the zoomed-in version of a specific part on the zig-zag path. In each subfigure, we demonstrate how the output probability of the sample we are interested in, say Xo, changes under the parameter update of each batch. The thin blue path is the update from other samples while the thick red path is the update from the Xo. For the red path specifically, the direction of this move is from the red dot to the red cross. The purple dot is q^t at the beginning of this epoch while the purple cross is where q^t is at the end of the epoch. As all the subfigures are on the same scale, the readers can directly read off why the zig-zag pattern for this sample occurs during training. The phenomenon described in Proposition 1 can also be clearly observed.
> >
> > >12. Proposition 1: $x_o$ has not been defined ...
> >
> > Thanks for pointing this out; we now define it.
> >
> > ## Continue in Part III

---

> > > ### Author Response · Authors · 2021-11-23
> > > **We answer most of the concerns in Appendix B,C, D. (Part III)**
> > >
> > > >13. Figure 4: What does the black dashed line...
> > >
> > > We intended that the left corner under the black dashed line was the region where argmax(p) is correct. But for simplicity, we remove such lines in this version.
> > >
> > > >14. Section 5.1: What labels are used for...
> > >
> > > Thanks for this question. Throughout the paper, the test performance is evaluated by a hold-out, clean OHT label dataset. We use the training set to train the network and the validation set to choose hyperparameters (including early stop criterion). Note that the training set and validation set always come from the same distribution (e.g., in section 5.2, the OHT labels of training and validation set are randomly flipped using the same strategy), and the label of the test set is always clean.
> > >
> > > >15.Section 5 & Figure 5: How does Filter-KD compare...
> > >
> > > As claimed by many related works (as with our analysis in sections 3-4), KD usually outperforms LS. Furthermore, the only difference between KD and ESKD is that we early stop the teacher's training in ESKD, hence we believe ESKD might be the strongest baseline among these 3 candidates. From our experiments, the performance of KD and LS are slightly lower than ESKD (but still better than OHT). Furthermore, as FilterKD also uses early stopping mechanisms, we think comparing it with ESKD and OHT is enough to verify that a network can spontaneously refine the label and FilterKD can reduce the high-variance issue in the ESKD case, that is why we leave these results out in the previous version.
> > >
> > > Regarding the optimum temperature, as claimed in Q1, we believe $\tau=1$ is a reasonably good choice for both FilterKD and ESKD.
> > >
> > > Anyway, to make the paper more persuasive, we added the results of KD in this version.

---

> > > > ### Comment · Reviewer_bjwd · 2021-11-24
> > > > **Response to author's rebuttal**
> > > >
> > > > **First and foremost, I want to praise the authors for their commitment and scientific spirit, as they considered and improved their submission based on most (if not all) of the comments in the reviews. Section 2.1 is now more succinct, figures are more readable and the appendix provides a comprehensive complement to the claims in the main paper. I also want to acknowledge the effort that goes into running a sizable set of additional experiments during the review process in order to assuage reviewers.**
> > > >
> > > > - In regard to the main weaknesses listed in my initial review:
> > > >   1. I see that the concept of noise has been re-worked and it is now clearer what the conditions of the problem are.
> > > >   2. Regarding the temperature, there are several claims on the appendix that are unscientific or inconclusive (e.g., analysis of the gradient, claims that the explanation becomes "awkward" or makes the paper "easier to follow", citing lack of consensus for a standard value) especially if the empirical results are affected. On the other hand, the second part of Appendix C does run, as one initially expects, a grid search for an optimal temperature and provides references supporting the use of a temperature with value 1. Unfortunately, the experiments (and related work) in appendix C do not seem to consider the case where labels are flipped, diminishing the applicability of these results to the set up in the main paper.
> > > >   3. Regarding the use of CIFAR10H, Appendix E provides unconvincing arguments to leave it out. First, the difficulty can be controlled indeed (as with the toy dataset) by adding noise to p_cifarh (in fact, this is what is done later in the appendix). Second, even though CIFAR10H may certainly not correspond to a perfect estimation of p*, it is shown to be a better one than the original OHT, and therefore, according to the paper's hypothesis, it should yield a better generalization. Most importantly, the fact that the results of the last two experiments in appendix E do not align with the main claims of the paper could mean that (a) annotations of CIFAR10H are not a good approximation of p* or (b) that the hypothesis in the paper can be falsified by the annotations of CIFAR10H. Note: I find that the analysis of the toy dataset is of great value. I strongly encourage you to keep all the analysis and experiments on that dataset. I originally meant to suggest that experiments on CIFAR10H should be included *in addition* to those reported on the toy dataset.
> > > >
> > > > ### Upon another full revision of the paper, I have two other observations to make:
> > > >   - For Section 3.2, we know that the optimization of the empirical risk will, on average, push all similar samples towards the same class, even if there are a few that are mislabeled. This is especially true for the toy dataset where the point density is relatively high. It is unnecessary to talk about "unknown forces" or "spontaneous refinement" if the optimization itself can already account for this phenomenon. Hard samples and their zig-zag pattern can simply correspond to "unfortunate" samples that are mislabeled but are otherwise easy to learn (or for which similar samples exist that are correctly labeled). In fact, this is the conclusion of section 3.3 but without all the formalisms. Section 3.3 makes a valid point but it clutters the storyline that is being built, making it more suitable for the appendix than for the main paper.
> > > >   - Section 4 moves away from the premise of the paper (a search for a better estimator of p*) and focuses exclusively on training with noisy labels. Ignoring all the theory up to this point, results from this section are interesting on their own. However, they are addressing a different problem, namely training under noisy labels. For these experiments, some straightforward comparisons can be done with [1,2,3].
> > > >
> > > > **Finally, even though I am impressed by the improvements that the authors have made to the original manuscript, I still find critical flaws in both the argument and the provided evidence that should support it. I am confident that in the short term, parts of this work can lead to valuable contributions in the field of ML, and I encourage the authors to pursue their inquiry with the same enthusiasm and dedication that they have shown during this submission process.**
> > > >
> > > > ### References:
> > > > [1] Cheng, Hao, et al. "Learning with Instance-Dependent Label Noise: A Sample Sieve Approach." International Conference on Learning Representations. 2020.
> > > >
> > > > [2] Kim, Youngdong, et al. "Nlnl: Negative learning for noisy labels." Proceedings of the IEEE/CVF International Conference on Computer Vision. 2019.
> > > >
> > > > [3] Han, B., et al. "Co-teaching: Robust Training of Deep Neural Networks with Extremely Noisy Labels." 32nd Conference on Neural Information Processing Systems (NeurIPS). 2018.

---

> > > > > ### Author Response · Authors · 2021-11-26
> > > > > **Response**
> > > > >
> > > > > Thanks again for your new comments and your engagement with our work!
> > > > >
> > > > > > Unfortunately, the experiments (and related work) in appendix C do not seem to consider the case where labels are flipped
> > > > >
> > > > > We've now run this experiment, doing a grid search for temperature values in the noisy-label setting (noise probabilities 0.1, 0.2, ..., 0.7). We can't update the paper anymore, but the takeaways are:
> > > > > - For every temperature, FilterKD > {ESKD, OHT}, with ESKD usually $\ge$ OHT.
> > > > > - The choice $T = 1$ is not always the best.
> > > > > We think this actually only strengthens the results of our paper; we'll include the versions with optimal temperatures in the main body in revision. Thanks for encouraging us to do this.
> > > > >
> > > > >
> > > > > > Regarding the use of CIFAR10H [...]
> > > > >
> > > > > On further reflection, we think you're right, and we plan to put CIFAR10H results into the main body in revision. We think that in particular Figure 17 (right) is particularly convincing as to the efficacy of this implicit label refinement via (Filter)KD: the final performance is better by using our procedure than by collecting expensive additional human annotations.
> > > > >
> > > > >
> > > > > > we know that the optimization of the empirical risk will, on average, push all similar samples towards the same class, even if there are a few that are mislabeled. [...]
> > > > >
> > > > > We agree that the overall message of Propositon 1 is perhaps not incredibly surprising. We think that writing the result relatively precisely in this way, however, makes it clear exactly what "similar samples" means (it's samples where the empirical tangent kernel is large) and precisely how the effect decays as the similar samples become classified more correctly. We think this helps put the story in section 3.3 (particularly Figure 4) on much firmer ground than simply appealing to intuition.
> > > > >
> > > > >
> > > > > > Section 4 moves away from the premise of the paper (a search for a better estimator of p*) and focuses exclusively on training with noisy labels.
> > > > >
> > > > > We disagree that the premise of the paper is "a search for a better estimator of p*". In our view, the point of the paper is more like pointing out that having better $p_\text{tar}$ can really help the final model. Section 4 helps advance that point by giving us a situation where we unambiguously know some points have large $\\lVert \\mathbf e_y - \\mathbf p^* \\rVert$, allowing us to study the effect of "hard" samples in training on this dataset. We can also try to estimate this using CIFAR10H, looking at $\\lVert \\mathbf e_y - \\mathbf p_{\text{CIFAR10H}} \rVert$; we will expect the takeaways to be similar, and in any case will add that exploration to the paper since it could be less confusing.
> > > > >
> > > > > In terms of comparison to algorithms for the noisy-label case: it's worth noting that many of these algorithms assume access to side-information that we don't need or use (e.g. certain points which are known to be "cleanly" labelled). Our primary goal of the paper isn't to propose FilterKD as a good algorithm for noisy-label settings, but rather to use the noisy-label setting as a "controllable" case where we expect to be able to get a better $p_\text{tar}$; we also see improvements even in the clean-label case.

---

> > > > > > ### Comment · Reviewer_bjwd · 2021-11-29
> > > > > > **On the remaining issues**
> > > > > >
> > > > > > After considering the latest response from the authors:
> > > > > >
> > > > > > > Our primary goal of the paper isn't to propose FilterKD as a good algorithm for noisy-label settings, but rather to use the noisy-label setting as a "controllable" case where we expect to be able to get a better; we also see improvements even in the clean-label case.
> > > > > >
> > > > > > This is an important point to highlight in the main paper for two reasons: (1) it sets expectations with respect to the problem, and positions FilterKD accordingly (i.e., not as a competitive method to do classification under noisy labels), and (2) it serves well as a segue between the methods and experiments.
> > > > > >
> > > > > > **Even though this re-interpretation of the noisy-label setting can be a slight stretch to get around evaluating SOTA on noisy-label classification, I think the scientific community can benefit from the contributions in this work, engage with its ideas, and ponder further on their value. I strongly advise the authors to include the aforementioned clarification in their paper. _Assuming they will_, I am more than happy to upgrade my initial assessment.**
> > > > > >
> > > > > > Finally, I would also like to extend my gratitude to the authors for an engaging and fruitful scientific discussion.

---

> > > > > > > ### Author Response · Authors · 2021-11-29
> > > > > > > **We will, thank you!**
> > > > > > >
> > > > > > > You’re right that we should definitely emphasize this point more in the paper, and will attempt to in the next version.
> > > > > > >
> > > > > > > Thank you as well for a discussion that has definitely benefited our work!

---

### Official Review · Reviewer_JaQp · 2021-10-31

**Correctness:** 2
**Technical Novelty And Significance:** 3
**Empirical Novelty And Significance:** 2
**Recommendation:** 5
**Confidence:** 3

**Main Review:**

The observed "zig-zag" pattern is interesting and gives one reason that early stopping is effective in knowledge distillation. The improvement obtained by early stopping and the proposed FilterKD is rather minor.

**Summary Of The Paper:**

This paper analyzes the learning process in knowledge distillation and proposes a method called Filter-KD. It was evaluated by using CIFAR10, CIFAR100, and AG News.

**Summary Of The Review:**

I am still not sure the proposed method is justified well in theory. It is based on NTK theory but it is applicable only for a sufficiently wide network. Also, several "zig-zag" patterns are shown in figures, but we are not sure of their definition and how often it appears in the real dataset. The improvement in accuracy is rather minor.

---

> ### Author Response · Authors · 2021-11-23
> **Response to reviewer JaQp**
>
> Thanks for your comments. Please see our [overall response](https://openreview.net/forum?id=Iog0djAdbHj&noteId=4nBozEauGKg), particularly Q2 about the scale of the practical improvement.
>
> In terms of the theoretical analysis: it's worth re-emphasizing that Proposition 1 actually does not make any assumption about the form of the network (other than differentiability). In an appropriate infinite-width regime, the term $\mathcal{K}^t$ is constant over the course of training, making it a little simpler to think about the meaning of Proposition 1. But the pattern we discussed still holds; the only difference is that the notion of "similarity", and hence which points pull the label towards them and which are ignored, can change over the course of training. We've clarified this a little further in the current revision.

---

### Official Review · Reviewer_QVYW · 2021-10-31

**Correctness:** 2
**Technical Novelty And Significance:** 3
**Empirical Novelty And Significance:** 3
**Recommendation:** 6
**Confidence:** 4

**Main Review:**

In terms of novelty, clarity, and significance, I think this paper is good. The question of why distillation improves performance is an important one. Analyzing the learning paths of individual examples is an interesting idea. To my knowledge, the visualization of learning paths is novel, as is the Filter-KD method. I enjoyed reading the paper, and I think the ideas described have the potential to inspire interesting future work.

With that said, I’m not entirely convinced that the explanation for the success of KD provided here is accurate for real-world KD use cases.

1. A major claim is that KD’s success is related to the zig-zag pattern, but this claim does not seem to be sufficiently supported by the experiments involving real data. It is shown that, on some examples, there is a zig-zag in the learning path, but there is no direct demonstration that this zig-zag is related to the efficacy of KD. One idea is to measure the degree of the zig-zag for each example and then verify that it suffices to use KD only for these examples and one-hot training for others. This seems like the bare minimum thing to show, but the entropy of the model distribution at the early stopping point might be correlated with the degree of zig-zag, in which case it seems somewhat trivial that the low-entropy outputs could be replaced with one-hot labels. I am not sure exactly what a good empirical test of the “zig-zag” hypothesis would look like, but I don’t find the evidence provided compelling enough.

2. It’s not clear to me whether distillation is performed at a temperature of 1 or a temperature that is tuned somehow. Assuming the magnitude of the final logits provides some indication of learning difficulty, I could see distillation at a temperature >1 providing some of the benefit of early stopping. This also seems like a relevant baseline for FilterKD, since both increasing the temperature and averaging predictions will have the effect of increasing the entropy of the target distribution.

3. The numbers in Table 1 seem uncompelling. The caption does not say whether the +/- is standard error, standard deviation, or something else, but assuming it is standard error, I don’t think the improvements on CIFAR-10 or AG News would be statistically significant. The improvements from KD on these datasets are small to begin with, so they may not be particularly good benchmarks for this work. The performance with noisy labels is more compelling, but doesn’t help to validate that the success of KD on clean data is attributable to the phenomenon described.

4. I’m suspicious of the idea of “base difficulty.” “Base difficulty” is defined as the L2 distance between the one-hot label and the true conditional distribution. A nice aspect of base difficulty is that it depends only on the true data distribution. However, it seems to capture only two kinds of “difficult” examples: those that are ambiguous (if the conditional distribution has high entropy) or incorrectly labeled (if the conditional distribution has low entropy). The proposed “integral difficulty” measure further captures the difficulty of _learning_ particular examples, and is somewhat similar to previously proposed metrics such as [C-score](https://arxiv.org/abs/2002.03206) (see also the learning speed based proxies in that paper, one of which is equivalent to integral difficulty) or [variance of gradients](https://arxiv.org/pdf/2008.11600.pdf). Intuitively, data points with low base difficulty could have high integral difficulty if they are located in regions of the distribution with low density. The labels in CIFAR-10/100 are relatively clean and the images are relatively unambiguous, so I suspect that most of the high integral difficulty examples do not actually have high base difficulty.

Minor:
- “Early-stopped knowledge distillation” is introduced without any further description. Only from the appendix is it clear that it means knowledge distillation from an early-stopped teacher, as opposed to early stopping during knowledge distillation.
- IIUC, the intuition from Section 3.3 is that the reason for the “zig-zagging” in Figure 3 is that the label assigned to a given point changes not only based on the gradient of the loss for that point, but also on the gradients of the losses on other similar points. This point seems intuitive even without bringing up the $\mathcal{A}^t$ or the empirical NTK, and I am not sure that Proposition 1 helps to understand it. Further analysis of the dynamics of $\mathcal{A}^t$ and the stationarity of the empirical NTK might make this decomposition more interesting, but the analysis in Appendix E doesn’t seem conclusive.
- Missing paren in definition of integral difficulty on p. 5.
- The idea of taking a moving average of network weights, as explored in Appendix G, is much older than MoCo and it would be good to cite relevant literature, e.g., [Ruppert](https://ecommons.cornell.edu/bitstream/handle/1813/8664/TR000781.pdf?sequence=1), [Polyak & Juditsky](https://epubs.siam.org/doi/abs/10.1137/0330046), and the [Inception paper](https://www.cv-foundation.org/openaccess/content_cvpr_2015/html/Szegedy_Going_Deeper_With_2015_CVPR_paper.html), which AFAIK was the first or among the first papers to apply this method in deep neural network training.

**Summary Of The Paper:**

This paper proposes an explanation for the success of distillation. It first experiments with synthetic Gaussian data. On synthetic data, it shows that distillation works better from an early-stopped model. It probes why, and finds that, when the one-hot label is far from the true conditional distribution, the early-stopped model’s distribution tends to be closer to the true distribution than either the one-hot label or the converged model. The early-stopped model is better because, over the course of training, the model tends to produce a distribution close to the true one before finally overfitting the one-hot label. Some examples show similar patterns for real-world networks/datasets, provided the predictions over training are smoothed. Motivated by these results, the paper proposes to use an averaged distribution of labels from different steps of training rather than the distribution at the end of training, and shows that this improves performance by a little bit.

**Summary Of The Review:**

This paper is well-written, novel, tackles an important question, and contains potentially interesting ideas. However, the evidence provided is not quite sufficient to validate the claims made. The suggestion that zig-zag in the learning trajectory is important to the success of distillation on real (non-noisy) data is not well-demonstrated, and the benefits of the proposed FilterKD technique seem marginal on real-world data.

---

> ### Author Response · Authors · 2021-11-23
> **Response to reviewer QVYW, part I**
>
> ## General Response:
>
> We thank the reviewer for pointing out these issues, which are very helpful in improving our paper. Please also note our [general response](https://openreview.net/forum?id=Iog0djAdbHj&noteId=4nBozEauGKg) above.
>
> The main concern of the reviewer is that the experiments in the previous version cannot persuade the readers that KD's success (especially in the clean label case) relates to the zig-zag pattern; we agree this case wasn't as strong as it could have been. In this version, we expand on their relationship, including more experiments.
>
> First, we want to highlight that KD's success (only considering learning from teacher's output) comes from the following two aspects:
>
> (a) better label (i.e., $p_\text{tar}$ is better than OHT label);
> (b) better training dynamics.
>
> The focus of our paper is (a). So we should not give the full credit of KD's success to the better label provided by the teacher. Aiming at (a), we want to find out how such a better label occurs. Hence we use the toy example and noisy-label CIFAR to illustrate the zig-zag pattern because the quality of their labels are easy to measure, and the label fixing process is also easy to observe (see Figure 2 and 6 in this version). We find that the network can spontaneously refine the label (i.e., making q move towards p*, or more directly, correct the wrong labels) during OHT training, and FilterKD can assist this.
>
> Regarding the clean-label real data case, we can only use test accuracy to indirectly prove KD is refining labels during teacher's training (maybe Figure 15 can also give some hints). Recall the loss function of KD (assuming the student only learn from the teacher, not OHT labels) and basic cross-entropy. When fixing $\tau=1$ (please refer to Appendix C for why we do that), the only difference among baselines is $p_\text{tar}$. Under such a condition, we believe the fact that FilterKD, ESKD > OHT (better in each run) is enough to conclude KD methods can provide better labels. Furthermore, as FilterKD is proposed to mitigate the high-variance issue in ESKD, and we can directly observe the zig-zagness of some ambiguous samples, we believe it is reasonable to make the conclusion in the paper.
>
> ## Piece-to-piece Part-I
>
> >Main review--> 1. A major claim is that KD’s success...
>
> First, we want to clarify that KD's success might come from both better labels and better optimization, but this paper only focuses on the label part. Regarding how zig-zag is related to better $p_\text{tar}$, and hence partially related to the efficacy of KD – this is a good question that we address in some detail in the general response.
>
> For the label refining part, we change the flow of the paper to highlight how we get to this conclusion:
>
> (1) From Hypothesis 1, we know better $p_\text{tar}$ leads to better test performance.
>
> (2) From toy data and CIFAR10 with 1000 flipped labels, we verify the network can spontaneously refine the label during OHT training. (Figures 6 and 2 are, we think, quite persuasive.)
>
> (3) Then, we experimentally find FilterKD > ESKD > OHT in the clean-label dataset. As the only difference among these methods is $p_\text{tar}$ (in each run, even the network initialization is the same), it is safe to claim Filter-KD can provide better  $p_\text{tar}$.
>
> Furthermore, from Figure 15, we can indeed observe some ambiguous samples with zig-zag paths, which indicates that FilterKD and ESKD indeed refine the labels (even when there are few bad labels) in the clean CIFAR case.
>
> Regarding the experiment mentioned by the reviewer, sorry for not getting to this in the time limit. We plan to try it in the future. But we believe the analysis and results in the current version are enough to claim FilterKD and ESKD can give better $p_\text{tar}$.
>
> >2. It’s not clear to me whether...
>
> Thanks for this question. Please refer to Q1 in the overall response and Appendix B for more discussions. By the way, changing the temperature affects both the labels used and the optimization process. So even if we achieve better performance with a higher temperature, we cannot conclude the enhancement only comes from the refined labels.
>
> >3. The numbers in Table 1 seem uncompelling...
>
> Thanks for pointing out this. The +/- was a standard error, but we now provide the detailed results for 5 runs in Table 6 (at the end of the paper), which shows the FilterKD outperforms other baselines in each run. Note that in each run, all the settings other than $p_\text{tar}$ are the same for different methods (optimizer, network initialization, etc). In terms of the small gap, please refer to Q4 in the overall response to all reviewers. For the role of noisy label example, please refer to section 4 in the paper. By the way, with fine-tuned hyper-parameters, we provide more compelling results in Table 2.
>
> ## Continue in part II

---

> > ### Author Response · Authors · 2021-11-23
> > **Response to reviewer QVYW, part II**
> >
> > >4. I’m suspicious of the idea of “base difficulty.”...
> >
> > We thank the reviewer very much for pointing out this problem. We totally agree with the reviewer on this point, i.e., a sample with high base difficulty is likely to have high integral difficulty, while the opposite is not true. Hence in this version, we change the flow of the paper and no longer use the term integral difficulty. We instead define a zig-zag score and provide some interesting discussions in Appendix C.
> >
> > Regarding the clean CIFAR, it is true that some unambiguous samples converge slowly due to the low-density reason (i.e., no similar samples that can "drag" it). But usually, the ambiguous samples converge even slower, as some similar samples might drag it to another direction. However, as we do not have the $p^*$ for these samples, we can only visualize samples with the highest zig-zagness to speculate what happened during training. By the way, we propose to use the integral of $q_{k\neq y}^t$ for the $k\neq y$ which maximizes that integral to measure the zig-zagness of the learning path; Figure 15 in Appendix C might be good evidence for this as a good score.
> >
> > In summary, we admit the explanation for the clean CIFAR case is hard and indirect, but we believe the reasoning process provided in the General Response part is enough for the results in the paper.
> >
> > > Minor --> 1. “Early-stopped knowledge distillation” is introduced...
> >
> > Thanks for pointing out this. We will clarify this in this version. By the way, to make a fair comparison, all OHT, LS, X-KD use early stopping during the student's training.
> >
> > > IIUC, the intuition from Section 3.3...
> >
> > We agree that the vague notion of "similar points matter" is intuitive, but think that Proposition 1 helps bring much more clarity to exactly what "similarity" means, and to confirm that this vague intuition is grounded in reality. In particular, we think it makes the explanation for the pattern in the new Figure 4 much more clear.
> >
> > > Missing paren in the definition...
> >
> > Thanks for pointing this out; this is resolved since we've replaced the notion of integral difficulty with the new zig-zag score.
> >
> > > The idea of taking a moving average...
> >
> > This is a very good point; we've updated the reference accordingly.

---

> > > ### Comment · Reviewer_QVYW · 2021-11-30
> > > **Response to authors**
> > >
> > > I thank the authors for their response, and I apologize for responding to them at the very end of this stage of the discussion period. My main concerns relate to point 1, which I still feel is not adequately addressed.
> > >
> > > In general, I'm not sure it's possible to distinguish the contributions from the labels with those from the training dynamics in a neural network. The changes made to the weights at each iteration involve an interaction between a learning rule, a loss function, and a set of labels, and we can only ask about what changing labels does in conjunction with the (potentially complicated) dynamics implied by the loss function and learning rule. If ESKD/FilterKD work because they provide objectively better labels, then we'd see labels that are closer to ground-truth when applying ESKD/FilterKD, but the converse is not true. Instead, it could be that ESKD/FilterKD provide labels that ease optimization, and labels that ease optimization on noisy data are objectively better in terms of distance to the true distribution, but the labels that ease optimization on clean data are not.
> > >
> > > FilterKD clearly has effects beyond correcting bad labels: if we consider two examples with the same base low difficulty, one of which is similar to many other training images and one of which is fairly unique, the former will generally be learned faster than the latter, because over the course of each epoch the network gets trained not just on that example but also on the similar examples, and thus the label for the latter will end up softer than the label for the former. Reducing the network's confidence in regions of the distribution that are likely to be seen less at test time could improve accuracy. However, I think that would actually be a different mechanism from the one suggested in the paper.
> > >
> > > I am reasonably convinced by the responses to the remaining points:
> > >
> > > 2. Changing the temperature is just changing the targets + scaling the student's output, so even if by "better labels" we mean labels that are better for SGD training, I'm not sure why one should lump temperature in with the things that change learning dynamics rather than things that change labels. With that said, from the authors' response to Reviewer bjwd, it seems like the authors have addressed this issue by adding experiments conducted with other temperatures, and I feel that this is a satisfactory solution provided that the claims in Appendix C get revised/removed.
> > > 3. I thank the authors for providing these numbers. I agree that the new results seem to demonstrate a significant advantage for FilterKD. I would encourage the authors to keep the AG News experiment from the previous revision in the paper, even if the advantage of FilterKD is not significant.
> > > 4. I agree that the zig-zag score is an improvement over the previous integral difficulty measure.
> > >
> > > Again, I think this paper performs some interesting analyses, but I still worry that there is a high probability that the explanation for the success of distillation that it provides is incorrect when it comes to real data, and thus I'm reluctant to change my score.

---

> > > > ### Author Response · Authors · 2021-11-30
> > > > **Thanks for the response, not too late.**
> > > >
> > > > Thank you for the response, which is definitely not too late; your concern is entirely legitimate, and we are also concerned about this. Optimization effects are, unfortunately, very difficult to disentangle from differences in terms of the labels themselves when dealing with deep networks. We plan to add some more detailed discussion and caveats on this point to a new revision, but we believe that based on the the current evidence, the explanation of the paper seems likely to be at least a significant factor in explaining the success of KD.
> > > >
> > > > > FilterKD clearly has effects beyond correcting bad labels [...]
> > > >
> > > > Thanks very much for providing this explanation; this indeed is a sensible reason for the success of FilterKD, especially in the clean-data case. We also observe some samples with a straight learning path that still have not converged at the teacher's early-stopping point. However, as in Figure 16, we indeed observe many zig-zag learning paths, suggesting that the mechanism mentioned in the paper indeed contributes to the final better performance. The zig-zag score we proposed in this version is also designed to separate these two cases: the sample mentioned by the reviewer might have a high integral difficulty but a low zig-zag score, because no similar samples "drag" it to be zig-zag.
> > > >
> > > > > Instead, it could be that ESKD/FilterKD provides labels that ease optimization [...]
> > > >
> > > > To answer this, we want to separate two highly related concepts:
> > > >
> > > > 1. why a student can perform better with these labels
> > > > 2. how a teacher can provide such better labels.
> > > >
> > > > We think the claim of the reviewer is asking 1, but our focus is 2. Better labels can surely provide better learning dynamics, potentially leading to better accuracy. Point 2 is entangled with point 1 and difficult to establish separately, but we believe the zig-zag effect explains 2 while still leaving some questions open in terms of 1.
> > > >
> > > > In summary, we want to highlight that we changed the flow of this paper somewhat:
> > > >
> > > > 1. we find NN can spontaneously refine labels;
> > > > 2. we take advantage of these mechanisms to get better labels;
> > > > 3. we verify with some experiments that we indeed get better labels.
> > > >
> > > > For 3, we fix all the settings other than $p_\text{tar}$  to make sure the improvement only comes from different labels. We don't fully understand, however, how those good labels affect the training process and finally give the student a better accuracy. The focus of the paper is to provide a way to "find these better labels" during the OHT training of the teacher, inspired by the zig-zag pattern and Hypothesis 1.
> > > >
> > > > The mechanism here does not fully explaining KD's success; temperature, teacher architecture, and other mechanisms all play roles which are perhaps related to the point of this paper, but not fully captured by it. This partial explanation is an interesting phenomenon of its own right, which we think we explain, and then we use to propose a new method. It is very possible (as you mention) that zig-zagging is not the only reason for FilterKD's success, but we believe it is an important aspect of it, based on the spread of evidence in the submission. We will clarify the discussion of this, including the possible mechanism you proposed, in the next revision.

---

> > > > > ### Comment · Reviewer_QVYW · 2021-12-01
> > > > > **I'm satisfied by this response**
> > > > >
> > > > > As I said in my initial review, I enjoyed reading this paper. I will raise my score with the understanding that the final paper will include additional discussion of the potential shortcomings of the provided explanation when it comes to clean data.

---

> > > > > > ### Author Response · Authors · 2021-12-01
> > > > > > **Thanks very much for your reply.**
> > > > > >
> > > > > > We thank the reviewer for the insightful comments. We will update the discussion and experiments about the clean data in the next version.

---

### Official Review · Reviewer_8bUF · 2021-11-01

**Correctness:** 3
**Technical Novelty And Significance:** 4
**Empirical Novelty And Significance:** 4
**Recommendation:** 8
**Confidence:** 4

**Main Review:**

Strengths:

1. The theoretical explanation and the illustrative toy experiments are very suggestive of what may actually be happening with ESKD. These contributions are very novel and useful in my opinion. They show that for harder examples, compared to easier examples, the learning path initially takes the network output to the true $\mathbf{p}^*$  before making a turn for the one-hot target.

2. The experiments when performed more real-world datasets using a proxy called "integral difficulty" instead of the true $\mathbf{p}^*$ (as it is unknown) still show very similar patterns, at least for the easy examples, compared to the toy Gaussian dataset where $\mathbf{p}^*$ can be calculated exactly. This is intellectually satisfying.

3. The experiments show how hard examples, as measured in terms of integral difficulty, show a very different -- highly erratic -- learning path compared to easier examples. More importantly, the authors show that by low-pass filtering, the path actually looks quite similar to the  case of medium hardness. This is very important as it allows the authors to propose an intuitive novel KD framework called FilterKD where the targets are smoothed teacher outputs.

4. The FilterKD algorithm is novel, intuitive relatively easy to implement for smaller datasets.

5. The experiments on CIFAR-10 and CIFAR-100 show that FilterKD is indeed better than using ESKD.

6. Experiments with noisy labels are well-performed and provide good evidence for the theoretical ideas as well as FilterKD.


Questions and suggestions for improvement:

Experimental results need to be improved. I would suggest the following additions

1. Results of regular KD in addition to ESKD.

2. There are results only with one teacher-student architecture pair. A wider set of teacher-student pairs is required before concluding that the proposed method is better.

3. In Table1, both the teacher and student have the same architecture. One of the main reasons to do KD is for model compression. So having a larger teacher and smaller student would be quite useful for a larger audience. I would suggest looking at the benchmark and code provided by Tian et al. in Contrastive Representation Distillation, ICLR 2020 as an easy starting point.

4. The memory requirements for storing smoothed targets in memory may be prohibitive. But why can't we load them from permanent memory as and when required, like we do with a dataloader for images and videos?

5. Experiments on a dataset like ImageNet or at least TinyImageNet would strengthen the paper more.

**Summary Of The Paper:**

Based on empirical observations and the neural tangent kernel theory, the authors present an explanation of why Knowledge Distillation (KD) and Early Stopping KD (ESKD) are better than using just the one-hot labels as well as how hard samples have high variance curve in the output space. Based on these observations, they propose a novel method called Filter-KD which yields better performance over using just one-hot labels as well as EKSD for standard vision datasets.

**Summary Of The Review:**

Overall, I believe there are solid contributions in this submission which can be further demonstrated with the suggested experiments. Hopefully, the authors will be able to address the weaknesses I have outlined.

For now, I am giving this a 6: marginal acceptance. I am happy to increase the rating based on the authors' response to the next level.

UPDATE AFTER AUTHOR RESPONSE:

I thank the authors for a clear response to my suggestions and the additional experiments. I am satisfied with the additions and modifications and I am increasing my score. Please try to add ImageNet results if possible, as it may help others in the field who use it as a common benchmark.

I have also read the other reviews and the authors have also addressed them well too.

---

> ### Author Response · Authors · 2021-11-23
> **Added some results, and will add more later.**
>
> Thank you for identifying the strengths of this work. We appreciate the thorough review and made some enhancements based on it. Due to the time limits, we cannot handle all of them, but we would update these experiments in the next version. If you have other concerns in mind, we would be happy to take them into consideration.
>
> In addition to the specific responses below, please also see [our general response](https://openreview.net/forum?id=Iog0djAdbHj&noteId=4nBozEauGKg).
>
> >Questions-->1. Results of regular KD in addition to ESKD.
>
> Thanks for pointing this out. As we find ESKD usually outperforms KD (the only difference between them is that we early stop the teacher's training in ESKD), we didn't record all the KD's results. However, to make the paper more persuasive, we provide the KD results in Tables 3 and 6. We would add KD to Figure 7, Table 1,2, as well in the next version.
>
> >2. There are results only with...
> >3. In Table1, both the...
>
> This is a good point. Due to the time limits (limited GPU time for several new experiments to add), we added the following settings on the CIFAR100 dataset in Table 2:
>
> EfficientB1—>ResNet18
>
> ResNet18—>MobileNetV2
>
> ResNet18—>VGG11
>
> All exhibit similar trends. We will try to add more thorough experiments in the latter version.
>
> >4. The memory requirements...
>
> Thanks very much for pointing out this solution. We might try this in our code. Actually, we also plan to reduce the precision of the variables for storing the probabilities to further reduce the memory cost. Considering the sparsity of those easy samples might also be a good way to make the algorithm more scalable.
>
> >5. Experiments on a dataset like ImageNet...
>
> Due to the time limits, we only add results on TinyImageNet using the setting of self-distill to EfficientNet-B0, under both clean and noisy cases. The results match our claims well.

---

> > ### Comment · Reviewer_8bUF · 2021-11-29
> > **Thank you for the response**
> >
> > I thank the authors for a clear response to my suggestions and the additional experiments. I am satisfied with the additions and modifications and I am increasing my score. Please try to add ImageNet results if possible, as it may help others in the field who use it as a common benchmark.

---

> > > ### Author Response · Authors · 2021-11-30
> > > **Thanks for your reply**
> > >
> > > Thanks very much for the reply. We would start to verify the results on ImageNet and add the results in the next version.

---

### Author Response · Authors · 2021-11-23
**Overall Response:**

We thank the reviewers for their helpful feedback, which we think has led to significant improvements in the new revision of the paper. As many reviewers had similar concerns, we highlight common issues here.

## Q0: Changing the flow of the paper (particularly sections 4 and 5)

In this version, we first clarify that refining the "bad" labels, which have a large distance to the ground truth $p^*$, can enhance the performance when we train a model. "Bad" labels can arise for multiple reasons, particularly the following two: first, a particular input might be ambiguous ($p^*$ has two or more classes with significant probability). Second, a relatively unambiguous input might have the "wrong" label in the training set.

We find the NN can spontaneously refine the bad labels during one-hot training by observing their learning path. This claim is explained by Proposition 1 and verified on the toy dataset (for ambiguous points) and CIFAR10 with 1000 labels randomly flipped (for "wrong" labels). Inspired by the CIFAR10 results, we point out the high-variance issue of ESKD's labels, and hence propose Filter-KD to solve it.

We want to especially highlight Figure 6 in the revision, which we think is the most persuasive experiment to show that the filtered learning path can indeed help us refine more bad labels: filtering does not simply choose a better point on the original path, it has fundamentally better predictions. We also add experiments with self-distillation on TinyImageNet, and distillation from different architectures, to show the superiority of Filter-KD.

Most importantly, we add Appendices C through E to answer the most important review questions.

## Q1: Temperature used in this paper

In Appendix C, we answer the following questions with a combination of analysis and experiments:

1. why we choose $\tau=1$,
2. why choosing $\tau=1$ still gives good baseline performance,
3. and how even for $\tau\neq1$, our analysis still holds – it just becomes a little more awkward.

## Q2: How typical the zig-zag pattern is (in both toy and real)

In appendix D, we use the following evidence to support that zig-zagness is quite common for the sample with a bad label:

1. We define a zig-zag score and show its correlation to base difficulty.
2. We use this score to verify that flipped labels in CIFAR10 indeed zig-zag more than other points.
3. We provide many more learning paths of random samples, for both noisy and clean CIFAR10, in Figures 15 and 16.

## Q3: Why not use CIFAR10H for proposition 1 and Figure 1/2/3?

We can use the label provided by CIFAR10H to verify hypothesis 1 (as illustrated in Figure 16), but there are various challenges detailed in Appendix E.

Replacing the toy dataset, it would also make the initial exposition harder to follow because the learning path is very high-variance (Figure 5) and a low-pass filter is needed to bring out the signal; we think the toy experiment is much clearer to bring across more or less the same overall points.

## Q4: Minor Improvement in Accuracy

First, we want to re-emphasize that the aim of this paper is not to pursue SOTA performance on CIFAR. With an extremely large network and pre-training on extra data, the SOTA performance of CIFAR10 (99.5 on paperswithcode) and CIFAR100 (96.1 on paperswithcode) are both very high (perhaps even over-fit). The aim of this paper is rather to study an interesting phenomenon during training, i.e., that neural networks can spontaneously refine the labels and FilterKD can assist in this refinement. We believe the consistent trend of FilterKD > ESKD > OHT supports this well.

Regarding the enhancement, it's worth noting that were building off a fine-tuned ResNet18 OHT baseline: 95.4 for CIFAR10 and 78.8 for CIFAR100. KD papers use quite inconsistent baselines, even for networks with similar numbers of parameters, for instance:

- CIFAR100:
    - Res20: 68.1 in [1]
    - Res18: 73.8 in [2]
    - Res18: 75.3 in [3]
    - Res20: 69.1 in [6]
- CIFAR10:
    - WideRes28: 93.31 in [4]
    - WideRes16: 93.48 in [5]

Furthermore, the labels in the CIFAR dataset are relatively clean. As the potential enhancement brought by FilterKD depends on how many "bad" training labels there are. More bad labels give the network more chance to spontaneously correct them. Also, as noted in Appendix B, we originally fixed many hyperparameters, especially temperature, – which have the potential to enhance ESKD and FilterKD's performance in our experiments – because we want to see how much improvement is directly caused by refining the labels. Under this setting, the FilterKD method still consistently outperforms others in each run.

In this version, we use better hyper-parameters (and a larger teacher) in Table 2. The enhancement is much larger (to our knowledge, the best-reported performance of ResNet18 on CIFAR100). However, we don't think we can owe this enhancement completely to the spontaneous label refining shown in the paper.

---

> ### Author Response · Authors · 2021-11-23
> **Reference part**
>
> [1]  A statistical perspective on distillation. Menon, Aditya K., et al. ICML 2021 http://proceedings.mlr.press/v139/menon21a.html
>
> [2] Refine Myself by Teaching Myself: Feature Refinement via Self-Knowledge Distillation. Ji, Mingi et al. CVPR 2021 https://openaccess.thecvf.com/content/CVPR2021/papers/Ji_Refine_Myself_by_Teaching_Myself_Feature_Refinement_via_Self-Knowledge_Distillation_CVPR_2021_paper.pdf
>
> [3] Self-distillation with Batch Knowledge Ensembling Improves ImageNet Classification. Ge, Yixiao et al. CVPR 2021[https://arxiv.org/pdf/2104.13298.pdf](https://arxiv.org/pdf/2104.13298.pdf)
>
> [4] Born again neural networks. Furlanello, Tommaso, et al.  ICML 2018 http://proceedings.mlr.press/v80/furlanello18a.html
>
> [5] On the efficacy of knowledge distillation. Cho, Jang Hyun, and Bharath Hariharan. CVPR 2019 https://openaccess.thecvf.com/content_ICCV_2019/papers/Cho_On_the_Efficacy_of_Knowledge_Distillation_ICCV_2019_paper.pdf
>
> [6] A popular repo: https://github.com/HobbitLong/RepDistiller

---

### Author Response · Authors · 2021-11-28
**Common letter to AC and reviewers**

Dear AC and reviewers:

We appreciate the comments, suggestions, and feedback thus far. You have helped us to improve the communication of the technical and empirical contributions in the submission.

We would be happy to follow up on post-rebuttal responses and look forward to addressing any outstanding points for discussion.

Please let us know if there are additional points of clarification that you would like us to address. We appreciate your time and energy through the review process and especially as we move into Discussion Stage 2 (even though it is the weekend and the discussion would end on Monday soon).

---

### Decision · Program_Chairs · 2022-01-20

**Decision:**

Accept (Poster)

**Comment:**

The authors made substantial  improvements to the originally submitted manuscript; however, reviewers initially remained reluctant to support the paper for acceptance based on the degree to which they were confident in the underlying arguments / position taken by the authors and the evidence provided to support their position and arguments. There are also concerns about the significance of the gains in performance afforded by the proposed approach.

During the author response period two reviewers became satisfied with the additions and modifications leading to an increase in the final score. It will be critical for the authors to try to add ImageNet results if possible in addition to other promises made to reviewers.

The AC recommends accepting this paper.